# Early-to-mid stage idiopathic Parkinson's disease shows enhanced cytotoxicity and differentiation in CD8 T-cells in females

Christophe M. Capelle[1,2,14], Séverine Ciré [1,15], Fanny Hedin[3], Maxime Hansen[4,5], Lukas Pavelka[4,5,6], Kamil Grzyb [4], Dimitrios Kyriakis [4,16], Oliver Hunewald [1], Maria Konstantinou[3], Dominique Revets[3], Vera Tslaf [1,2,6], Tainá M. Marques [6], Clarissa P. C. Gomes [4], Alexandre Baron[1], Olivia Domingues[1], Mario Gomez[3], Ni Zeng [1,2], Fay Betsou[7,8], Patrick May [4], Alexander Skupin [4,9,10], Antonio Cosma [3], Rudi Balling [4,11], Rejko Krüger[4,5,6], Markus Ollert [1,12] ✉ & Feng Q. Hefeng [1,13] ✉

Neuroinflammation in the brain contributes to the pathogenesis of Parkinson's disease (PD), but the potential dysregulation of peripheral immunity has not been systematically investigated for idiopathic PD (iPD). Here we showed an elevated peripheral cytotoxic immune milieu, with more terminally-differentiated effector memory (TEMRA) CD8 T, CD8+ NKT cells and circulating cytotoxic molecules in fresh blood of patients with early-to-mid iPD, especially females, after analyzing > 700 innate and adaptive immune features. This profile, also reflected by fewer CD8+FOXP3+ T cells, was confirmed in another subcohort. Co-expression between cytotoxic molecules was selectively enhanced in CD8 TEMRA and effector memory (TEM) cells. Single-cell RNA-sequencing analysis demonstrated the accelerated differentiation within CD8 compartments, enhanced cytotoxic pathways in CD8 TEMRA and TEM cells, while CD8 central memory (TCM) and naïve cells were already more-active and transcriptionally-reprogrammed. Our work provides a comprehensive map of dysregulated peripheral immunity in iPD, proposing candidates for early diagnosis and treatments.

Parkinson's disease (PD) is the second-most common neurodegenerative disease after Alzheimer's disease (AD), affecting approximately 10 million people worldwide[1,2]. In addition to the neuron-autonomous mechanisms, microglial activation and neuroinflammation in the brain of PD patients are also implicated in the pathogenesis of PD[3]. At the same time, PD patients are characterized by altered levels of several circulating cytokines[4,5]. Moreover, targeted investigations of selected immune cells in the peripheral blood found a reduction of CD4 T cells in PD patients vs healthy controls[6]. Not only total CD4 T cells, but also specific CD4 subsets, such as CD4 regulatory T cells (Treg) and Th17 have shown a reduction[7], although Th17 have been observed enhanced

in PD patients by another study[8]. Thus, the role of CD4 subsets in the PD pathogenesis is still controversial[9].

Emerging evidence suggests the involvement of peripheral CD8 T cells in other neurodegenerative diseases, e.g., AD[10]. In PD, cytotoxic CD8 T-cell infiltration has been reported in post-mortem brain tissues even before the α-synuclein (α-syn) aggregation and neuronal death, suggesting a potential role of CD8 T cells in initiating PD pathology[11]. α-syn-specific T cells have also been reported in the peripheral blood of PD patients[12] and associated with pre-clinical and early PD[13]. In a genetic PD mouse model, mitochondria-antigen-specific CD8 T-cell responses have been shown in both the periphery and brain[14]. Another

PD familial gene, LRRK2, regulates inflammation during infection of an animal model[15]. We have recently shown that another key familial PD gene, DJ-1/PAKR7, also acts on T-cell compartments by regulating immunoageing[16], as demonstrated in both patients and old mice[17]. However, it still remains unknown whether any specific subsets of T cells and other peripheral immune cells contribute to the pathogenesis of idiopathic PD (iPD), i.e., PD without a defined genetic cause. Since most PD cases are idiopathic, it is important to identify non-genetic factors contributing to iPD, e.g., peripheral immune dysregulations.

Although one immune cell type rarely works alone[18,19], most of the aforementioned studies in PD have so far focused on a few selected peripheral immune subsets following predefined hypotheses. Those hypothesis-driven studies cannot identify previously-unrecognized changes. Although a recent study has used unbiased single-cell RNA-seq analysis[20], their analysis was constrained to T cells. Furthermore, their conclusions were compromised by a single-digit number of PD patients and the lack of disease stage information. Meanwhile, the community is still seeking for easily-accessible, validated peripheral cellular or molecular biomarkers to enable for the early diagnosis of PD[21]. To address these questions, we applied a systems-immunology approach to comprehensively analyze the peripheral immune system of early-to-mid stage PD patients. By focusing on early-to-mid stage patients, we reasoned that we could have a higher probability to identify the disturbed peripheral immune cells initiating the pathology of iPD, rather than those responding secondarily to the manifestation of iPD pathological events.

## Results

### Early-to-mid stage PD shows a more-cytotoxic & late-differentiated immune profile

In the discovery analysis, we systematically scrutinized various immune subsets and their functional states in 28 PD patients (25 iPD aged 60-70 years and three genetic PD patients with mutations in GBA or PINK1) and 24 matched healthy controls (HC) (refer to "cohort design" in "Methods", Supplementary Tables 1 and 2; for simplicity, 'PD patients' are hereafter shortened as PD). This was realized by investigating 37 different innate and adaptive immune subsets and more than 700 combinatorial T-cell features, using a 35-marker mass cytometry (also known as Cytometry by Time of Flight or CyTOF) panel and five panels of multiple-color flow-cytometry (FCM) composed of 33 lineage and functional T-cell markers (Fig. 1a, Supplementary Tables 3 and 4). We recruited the participants from the ongoing nation-wide Luxembourg Parkinson's study with more than 800 PD and 800 HC[22] (https://www.parkinson.lu/research-participation) and controlled for several major confounding factors, cytomegalovirus (CMV) serostatus, medications and comorbidities, known to affect the immune system, to ensure that our observations are PD-specific (Fig. 1a and Supplementary Table 1). Furthermore, we narrowed the patients to those with early-to-mid stage disease [Hoehn and Yahr (H&Y) staging scale: mean = 2.3, ranging from 1.5 to 3.0; most of them ≤2.5, except for five participants with a scale of 3]. Most of the selected patients had a disease duration within 10 years from clinical diagnosis while three of them had a disease duration of 12, 13 and 19 years, respectively.

As a strong cryopreservation effect has been observed on the FCM readouts of T-cell markers[23], we performed the cytometric analysis on fresh blood (Supplementary Fig. 1). Our CyTOF analysis did not show a different general immunological fingerprint of PD vs HC, based on the entire peripheral immune system (Fig. 1b). Nevertheless, several immune cell types, especially the T-cell compartments, were altered (Fig. 1c). Total classical αβ T cells were modestly reduced in PD (Supplementary Fig. 2A), reflected by a decrease of total CD4 T cells (Supplementary Fig. 2B), whereas the γδ T cells were unchanged among total living CD45+ cells (Supplementary Fig. 2C). The decreased

frequency of total CD4 T cells was mainly due to a reduction in CD4+CXCR5+ T follicular helper cells (Tfh) (Supplementary Fig. 2D), CD45RA+CCR7+ naïve (Supplementary Fig. 2E) and CD45RA-CCR7+ central memory (TCM) CD25- conventional T cells (Tconv) (Supplementary Fig. 2F), but not CD45RA-CCR7- effector memory (TEM) Tconv (Supplementary Fig. 2G). Although total CD8 T cells showed no difference in PD vs HC (Fig. 1c, d), the CD8 naïve/memory subset composition displayed alterations (Fig. 1c). Our unbiased volcano plot analysis (Fig. 1c) showed that the frequency of cytotoxic terminally-differentiated effector memory T cells (CD45RA+CCR7-, TEMRA)[24,25] was substantially increased among total CD8 T cells (Fig. 1c, e), whereas the frequency of CD8 TCM was reduced in PD (Fig. 1f). The proportion of CD8 naïve and TEM showed similarity between PD and HC (Supplementary Fig. 2H, I). Furthermore, the expression levels (median signal intensity, MSI) of CD57, a marker for terminal differentiation, among CD8 TEMRA showed a trend to be increased (p = 0.0663) in PD (Fig. 1g). A trend to be increased was also observed for the frequency of CD57+ cells among CD8 TEMRA (p = 0.0568) (Supplementary Fig. 2J), further indicating a late-differentiated CD8 T-cell profile. Moreover, another cytotoxic cell type, natural killer T cells (NKT) also exhibited a late-differentiated state, as reflected by an increased frequency of CD8+ NKT (Fig. 1h), among total NKT[26], especially in women. Meanwhile, the frequency of less-differentiated CD4+ (Fig. 1i) and CD4-CD8- (also known as double negative, DN, Fig. 1j) NKT was either decreased or intact, respectively. The female-biased relative increase in PD might be explained by a lower fraction of CD8+ NKT in healthy females than males in their sixties (Supplementary Fig. 2K). CD8+ NKT also expressed higher levels of CD57 (Fig. 1k). The frequency of CD56highCD57- immature NK was also decreased (Fig. 1l). As CD8 T-cell composition was considerably changed, while total CD8 T cells were intact, we performed an unsupervised analysis on gated CD8 T cells. Indeed, our unsupervised viSNE analysis confirmed an enhanced frequency of CD8 TEMRA (CD45RA+CCR7-CD27-) (Fig. 1m). In brief, we observed a more-cytotoxic and late-differentiated immune profile in early-to-mid stage PD, as reflected by several relevant subsets, such as CD8 TEMRA, NKT and NK.

With the 35-marker CyTOF analysis in whole blood, we could assess many other immune subsets, such as granulocytes (neutrophils, eosinophils and basophils), monocytes (classical, intermediate and non-classical), dendritic cells (myeloid DC and plasmacytoid DC, known as mDC and pDC respectively), NK (immature and late), B cells (naïve, memory, plasma cells) and innate lymphoid cells (ILCs: ILC1, ILC2 and ILC3) (Supplementary Fig. 1A–M and Supplementary Table 5). Most of them did not show any significant change in PD vs HC regarding the frequency among total CD45+ cells or among the relevant parent gates (Supplementary Table 5). Consistent with a more-cytotoxic profile, an increased frequency of neutrophils was observed in PD, especially for males (Fig. 1n, Supplementary Fig. 2L). The heightened frequency of neutrophils was accompanied with a reduction in eosinophils, especially for males (Fig. 1o, Supplementary Fig. 2M). Unlike the scenario of CD8+ NKT, the gender-biased changes in these two types of granulocytes in PD cannot be simply explained by the pre-existing difference between male and female HC (Supplementary Fig. 2L, M). In the meantime, basophils were unchanged in PD vs HC (Supplementary Fig. 2N). The reduced eosinophils and increased neutrophils are conforming to two recent works[27,28], respectively, where although mainly routine whole blood counts have been analyzed.

Unexpectedly, the frequency of ILC2[29] in PD was decreased to almost half of that in HC (Fig. 1p, q), while ILC1 and ILC3 showed no difference (Supplementary Fig. 2O, P). The reduction was even more pronounced in males, which could be explained by a higher number of ILC2 already in male vs female HC in their sixties (Supplementary Fig. 2Q). The reduction in ILC2 is consistent with the observed decrease in eosinophils, as ILC2 control eosinophil homeostasis at

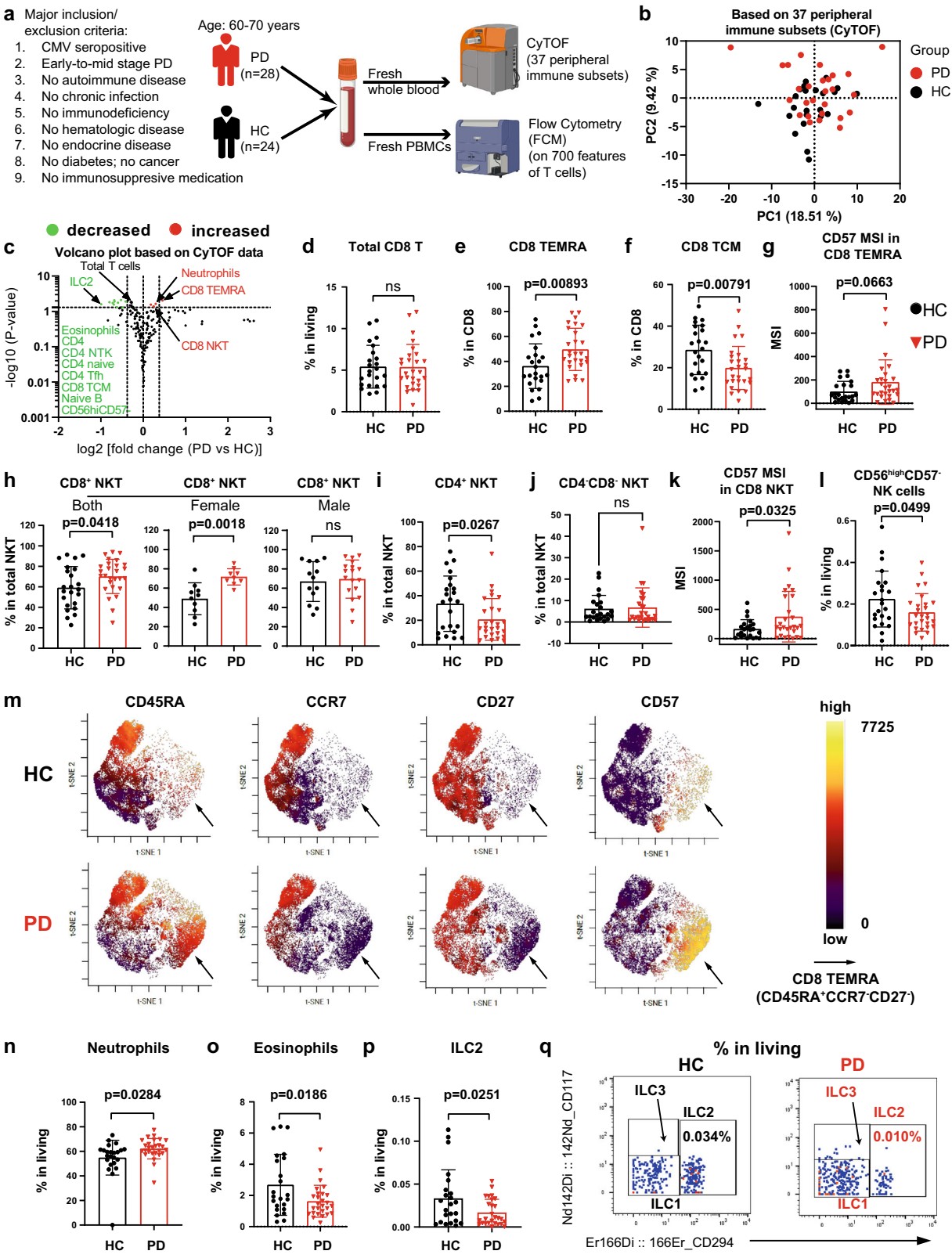

least in typical Type II immunopathology (i.e., allergy)[30]. Finally, we found a slight but significant decrease in the frequency of IgD+CD27− naive B cells (Supplementary Figs. 1 and 2R). Collectively, our unbiased comprehensive immunophenotyping analysis reveals a more-cytotoxic and late-differentiated immune profile, while the frequency of immature NK, eosinophils and ILC2 was decreased in early-to-mid stage PD.

### Early-to-mid stage PD exhibits a heightened effector profile in CD8 T cells

Based on our T-cell-dominant observations by CyTOF, we next analyzed T cells in depth using five FCM panels with a total of 33 T-cell-relevant markers (Supplementary Table 4), the combinations of which gave rise to ~700 features. The PCA determined a distinct immunological fingerprint in iPD vs HC (with the exception of one PD and one HC

**Fig. 1 | Single-cell CyTOF analysis shows a more-cytotoxic and late-differentiated immune profile in early-to-mid stage iPD. a** Graphical representation of the cohort and experimental setup. **b** PCA plot showing no distinct immunological fingerprint of PD based on the entire peripheral immune system. **c** Volcano plot highlighting the most significantly (p < 0.05, fold change >1.3) decreased and increased subsets in PD vs HC. The horizontal dashed line corresponds to −log10(0.05), while the two vertical dashed lines correspond to log2 value of −0.3785 or 0.3785. The marker definition for some of the highlighted subsets is provided here or other panels: CD4 naive (CD45RA⁺CCR7⁺), CD4 Tfh (CD45RA⁻CXCR5⁺) and naive B (IgD⁺CD27⁻) (Supplementary Fig. 1). Scatter dot plots showing the frequency of total CD8 T cells (**d**), CD8 CD45RA⁺CCR7⁻ (the simplified gating for TEMRA) (**e**), CD8 TCM (CD45RA⁻CCR7⁺, central memory) (**f**) and CD57 MSI in CD8 TEMRA (**g**). Scatter dot plots showing the frequency of CD8⁺ NKT (CD8⁺CD4⁻CD3⁺CD19⁻CD56⁺TCRgd⁻, **h**), CD4⁺ NKT (CD8⁻CD4⁺CD3⁺CD19⁻CD56⁺ TCRgd⁻, **i**) and CD4⁻CD8⁻ (DN) NKT (CD8⁻CD4⁻CD3⁺CD19⁻CD56⁺TCRgd⁻, **j**) and CD57 MSI in total NKT (**k**). **l**, Frequency of CD56ʰⁱCD57⁻ immature NK cells.

**m** Representative viSNE plot from either HC or PD highlighting the expression levels of CD45RA, CCR7, CD27 and CD57 in total CD8 T cells. The arrow indicates the area of CD8 CD45RA⁺CCR7⁻CD27⁻ (TEMRA). Scatter dot plots showing the frequency of neutrophils (CD66b⁺CD45ᵐⁱᵈCD16⁺CD294⁻, **n**), eosinophils (CD66b⁺ CD45ᵐⁱᵈCD16⁻CD294⁺, **o**) and innate lymphoid cell type 2 (ILC2, **p**). **q** Representative cytometry plots showing the expression of CD294 and CD117. The number showing the frequency among living CD45⁺ singlets. The results in (**c**–**l**, **n**–**p**) were analyzed using unpaired two-tailed Student's *t* test without adjustments for multiple comparisons. Data are presented as mean ± standard deviation (s.d.). Each symbol represents the measurement from one individual (**d**–**l**, **n**–**p**). ns or unlabeled, not significant; all significant *P*-values are indicated. CyTOF mass cytometry, CMV cytomegalovirus, NKT Natural killer T cells, PBMC peripheral blood mononuclear cells, HC healthy controls, *n* = 24, PD patients with Parkinson's disease, *n* = 28; MSI Median signal intensity. Panel **a** was created with BioRender.com. Source data are provided as a Source Data file.

labeled as "PD9" and "HC17" in Fig. 2a). The three genetic PD were not identified as outliers compared to iPD in the PCA plot based on the comprehensive T-cell analysis. Alike to the CyTOF data, no difference was observed in the frequency of total CD8 T cells among fresh peripheral blood mononuclear cells (PBMC) in iPD vs HC (Supplementary Fig. 3A). Different from the CyTOF data, we did not observe any significant difference in the frequency of total T cells and CD4 T cells between the two groups (Supplementary Fig. 3A). This might be due to the exclusion of granulocytes (accounting for the majority of immune cells in whole blood) in PBMC. Although the overall frequency of major T-cell populations remained unchanged in the FCM analysis (Supplementary Fig. 3A), the PCA results indicated the existence of changes in specific T-cell subsets (Fig. 2a).

Among the most significantly changed immune subpopulations, the FCM analysis again found a strong increase in the frequency of TEMRA among total CD8 T cells (Fig. 2b, c), confirming our CyTOF results. CD8 TEMRA cells re-express CD45RA, but lose the expression of CD45RO, CCR7 and CD27. Following a simplified gating strategy, we were able to pinpoint that CD45RA⁺CD45RO⁻CCR7⁻ cells (CD8 TEMRA) were increased among total CD8 T cells in iPD (Fig. 2c). To more strictly identify CD8 TEMRA, we next included CD27 in another staining panel. Similar to that from the simplified gating strategy (Fig. 2c), the frequency of CD45RO⁻CCR7⁻CD27⁻ effector CD8 T cells was also increased (Supplementary Fig. 3B). As the fraction of CD45RA/CD45RO double-negative (mean: ~5%) or double-positive (mean: ~5%) cells was tiny among CD8 T cells (Supplementary Fig. 3C), most of the CD45RO⁻CCR7⁻CD27⁻ effector CD8 T cells (Supplementary Fig. 3B) should be CD45RA⁺CCR7⁻CD27⁻ CD8 TEMRA, and thus also referred as CD8 TEMRA in certain analyses. Meanwhile, TCM (CD45RO⁺CCR7⁺CD27⁺) and transitional memory (TM) (CD45RO⁺CCR7⁻CD27⁺) CD8 T cells were reduced (Supplementary Fig. 3D, E), while naive (CD45RO⁻CCR7⁺CD27⁺) CD8 T cells showed no difference between PD and HC (Supplementary Fig. 3F). The reduction in CD8 TCM and TM cells was in line with a lower frequency of long-lived memory (KLRG1⁻CD127⁺)[31] CD8 T cells (Supplementary Fig. 3G) and consistent with the CyTOF results (Fig. 1f). Although total CD8 cells were not changed, the mean ratios of CD8 TEMRA over TCM were substantially enhanced from 12.05 in HC to 42.90 in iPD (Fig. 2d). Jointly, we firmly demonstrate that the portion of CD8 TEMRA is increased while the fraction of CD8 TCM and TM is contracted in the peripheral blood of early-to-mid stage iPD, indicative of an imbalance in the differentiation process within CD8 compartments.

We next investigated the functional states of various T-cell subsets. T-bet is an essential marker for effector CD8 lymphocyte functions[32,33]. Consistently, CD8 T cells from iPD vs HC also displayed a higher frequency of T-betʰⁱᵍʰ and CD45RO⁻T-bet⁺ CD8 T cells (Fig. 2e, f). T-bet⁺ cells were mainly CD45RO⁻, but not CD45RO⁺ cells, indicating

that those cells were mostly CD45RA⁺ terminally differentiated cells, based on the largely mutually exclusive relationship between CD45RA and CD45RO expression (Supplementary Fig. 3C). The increased CD8 effector function was further consolidated by an augmented proliferation and activation levels among CD45RO⁻ CD8 T cells, as quantified by the expression of Ki67 (Fig. 2g) and HELIOS[34] (Fig. 2h), respectively. The increase of CD45RO⁻CD57⁺ (Fig. 2i) and CD57⁺ (Supplementary Fig. 3H) portions among CD8 T cells further supports an advanced differentiation state[35] of CD8 T cells in iPD.

We further analyzed whether CD8 T cells from iPD show any signs of exhaustion[36], senescence[37] or other major dysfunctionality, as our participants were mainly over 60. We did not observe any sign of exhaustion in the expression of several key T-cell exhaustion markers such as PD-1, CTLA4 and LAG3 (Supplementary Fig. 3I–K). Moreover, the CD8 T-cell senescence marker KLRG1 was also unchanged between iPD and HC (Supplementary Fig. 3L). The activation marker ICOS was significantly decreased in CD8 T cells (Supplementary Fig. 3M). However, this decline only reflected the decrease in the frequency of CD8 TCM cells (Supplementary Fig. 3D), as only the frequency of ICOS⁺CD45RO⁺, but not ICOS⁺CD45RO⁻ cells was lessened (Supplementary Fig. 3N). We also observed a decreased expression of the amino acid transporter CD98 (Supplementary Fig. 3O), nevertheless being independent of CD45RO expression levels (Supplementary Fig. 3P). Together, these data support that CD8 T cells in early-to-mid stage iPD exhibit a terminally-differentiated but non-exhausted and non-senescent state.

Effector CD8 T cells tend to migrate to the tissues where an active immune response occurs[38]. Therefore, we also assessed the expression of several lymphocyte-trafficking relevant chemokine receptors, such as CXCR3, CCR4 and CCR6. In the blood of iPD, we observed a lower frequency of CXCR3⁺ and CCR4⁺ cells and their corresponding expression levels (geometric mean, MFI) among total CD8 T cells, whereas CCR6 showed no difference (Supplementary Fig. 4A, B). The decrease in CXCR3 and CCR4 expression might be attributable to the decline in the frequency of TCM cells among CD8 T cells (Supplementary Fig. 3D), since among all the four subsets, CD8 TCM cells displayed the highest expression levels of these two chemokine receptors (Supplementary Fig. 4C). The expression of the three individual receptors on CD8 TEMRA was low relative to that on other CD8 memory subsets, but without showing clear difference in iPD vs HC (Supplementary Fig. 4C). The frequency of CCR4 and CCR6 co-expressing cells was also decreased in iPD (Supplementary Fig. 4D), although the cells expressing both receptors was sparse among CD8 T cells (<1% in average). Accordingly, the frequency of cells lacking all the tested three chemokine receptors (CXCR3, CCR4 and CCR6) was increased in iPD vs HC (Supplementary Fig. 4D). We further asked whether CD8 TEMRA express one of the major homing receptors allowing them to properly migrate to the central nervous system (CNS). As integrin alpha 4 (also known as CD49d) is the major brain

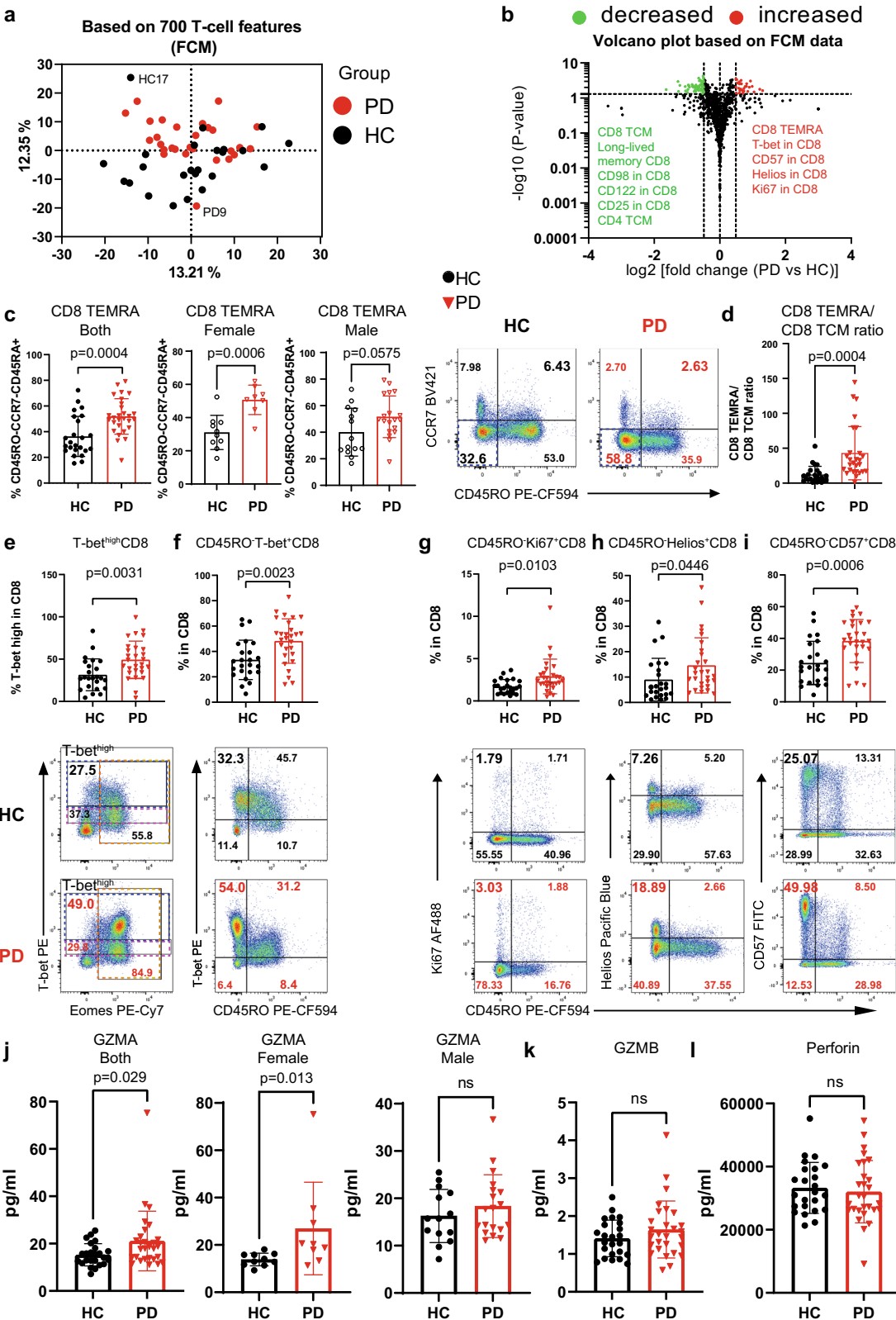

homing factor of peripheral CD8 T cells, controlling trafficking of CD8 T cells into the CNS[39,40], we analyzed CD49d expression. However, we did not find any significant difference in CD49d expression or the frequency of CD49d⁺ cells between iPD and HC among various subsets of CD8 T cells, including CD8 TEMRA (Supplementary Fig. 4E, F). In brief, our data suggest that CD8 TEMRA have an uncompromised potential to migrate into the CNS.

Since CD8 T cells and NKT are known to have cytotoxic potential and also secrete cytotoxic effector molecules, we asked whether a more-cytotoxic state was already reflected by serological parameters. To this end, we measured two of the most abundant extracellular cytotoxic effector molecules, granzyme A (GZMA) and granzyme B (GZMB), together with the membrane-pore forming molecule perforin. Highly encouragingly, circulating GZMA was significantly elevated in

**Fig. 2 | Early-to-mid stage iPD exhibits an increased effector profile in CD8 T cells. a** PCA plot showing a distinct immunological fingerprint of PD vs HC based on T-cell combinatorial features using the FCM analysis. **b** Volcano plot showing the most significantly ($p < 0.05$, fold change >1.4) decreased and increased sub-populations in PD vs HC. The horizontal dashed line corresponds to the value of 1.3 ($p = 0.05$), while the two vertical dashed lines correspond to the log2 value of −0.485 or 0.485 (fold change of 1.4). **c** Scatter dot plots (left) and representative FCM plots (right) showing the increase in CD8 TEMRA (CD45RO⁻CD45RA⁺CCR7⁻, the simplified gating strategy for TEMRA) for all (left) or females (middle) or males (right). The TEMRA gate was highlighted in blue dashed rectangle in FCM plots. The combination of markers used to define TEMRA was described in the y-axis title. **d** Scatter dot plots showing the ratios between CD8 TEMRA and CD8 TCM (CD45RO⁺CCR7⁺CD27⁺). **e-i** Scatter dot plots (upper) and representative FCM plots (lower) showing the frequency of T-bet^high (**e**), CD45RO⁻T-bet⁺ (**f**), CD45RO⁻Ki67⁺

(**g**), CD45RO⁻Helios⁺ (**h**) and CD45RO⁻CD57⁺ (**i**) among CD8 T cells. In **e**, blue and magenta dashed rectangle highlights T-bet^high and T-bet^mid cells, respectively, while gold dashed rectangle shows Eomes⁺ cells. The percentage of the interest was enlarged. **j-l** Serological levels of GZMA (**j**) for all (left) or females (middle) or males (right), GZMB (**k**) and perforin (**l**). The results in **b-i** were analyzed using unpaired two-tailed Student's *t* test while the results in **j-l** were analyzed using unpaired two-tailed Mann–Whitney nonparametric test without adjustments for multiple comparisons. Data are presented as mean ± standard deviation (s.d.). Each symbol represents the measurement from one individual (**c-l**). ns or unlabeled, not significant; all significant *P*-values are indicated. HC healthy controls, $n = 24$; PD, patients with Parkinson's disease, $n = 28$; FCM flow cytometry; GZMA/GZMB Granzyme A/B, TCM central memory. For **c**, **d**, **j**, female HC, $n = 10$; female PD, $n = 8$; male HC, $n = 14$; male PD, $n = 19$. Source data are provided as a Source Data file.

iPD, especially in females (Fig. 2j), while the levels of both GZMB and perforin were comparable between iPD and HC (Fig. 2k, l). In short, our serological data also suggest a more-cytotoxic immune milieu in iPD.

Although naive CD4 cells displayed no difference, the frequency of CD4 TCM cells was slightly, but significantly decreased (Supplementary Fig. 5A, B). Moreover, the portion of intermediate CD4 (CCR7⁻CD27⁺CD45RO⁻)[41] T cells was increased in iPD (Supplementary Fig. 5C). Unlike CD8 T cells, the frequency of effector and TEMRA among total CD4 T cells remained unchanged (Supplementary Fig. 5D, E). Together, CD8, but not CD4 T cells, favor a terminally-differentiated effector program over the generation of a long-lived central memory T-cell profile in iPD.

The combination of the chemokine receptors, such as CXCR3, CCR4 and CCR6 can also be used to distinguish the CD4 T helper lineages Th1, Th2 and Th17 (Supplementary Fig. 5F). By analyzing the combinations of these chemokine receptors and also assessing the expression of the CD4 Th master transcription factors T-bet, GATA3 and RORγT, we observed neither a significant change in the frequency of Th subsets, nor in the ratios between those subsets in iPD vs HC (Supplementary Fig. 5G–J). No change in any Th subset was also in agreement with our CyTOF results (Supplementary Table 5). We also did not observe any significant difference in the Th1 and Th2 cytokines, such as GM-CSF, IFN-γ, IL-5 and IL-13 measured in the sera of our cohort (Supplementary Fig. 5K). Furthermore, since IL-33 plays a crucial role in regulating the ILC2 response and serves as an alarmin in CNS[42], we quantified the circulating IL-33 level and found it was similar between PD and HC (Supplementary Fig. 5K). We further measured several CNS-homing relevant chemokines, such as IL-8, IP10 and MCP-1. None of them showed any significant difference (Supplementary Fig. 5K). Together, our data from various viewpoints firmly suggest that the CD4 composition is less affected in early-to-mid stage iPD.

### Early-to-mid stage PD shows a reduced frequency of CD8 Treg but not CD4 Treg
CD4 regulatory T cells (CD4 Treg) play an important role in suppressing effector T-cell responses to avoid an overshooting immune reaction that could harm surrounding self-tissues[43–45]. Previous reports have shown a reduced frequency of CD4 Treg and/or an impaired suppressive capability of Treg in PD and other neurodegenerative diseases[7,46,47]. Through our FCM analysis, however we did not observe any significant change in the frequency of CD4⁺FOXP3⁺ T cells (CD4 Treg) among CD4 T cells between PD and HC (Fig. 3a). Unexpectedly, we found that the frequency of CD8⁺FOXP3⁺ Treg (CD8 Treg) was reduced in iPD, especially for females (Fig. 3b, c). CD8 Treg are able to suppress the responses of effector cells, including CD8 cells [reviewed elsewhere[48,49]]. In line with our observation of reduced CD8 Treg, the expression of other markers relevant to CD8 Treg, such as CD25 (IL2RA) and CD122 (IL2RB)[50], was also reduced in CD8 T cells of iPD vs

HC (Fig. 3b). The ratio between the frequency of CD8 TEMRA and of CD8 Treg was increased, with a mean of 70.62 in PD vs 30.37 in HC, further highlighting an effector-dominant, and consequently more 'active' CD8 T-cell compartment, especially for females (Fig. 3d). Although the frequency of CD4 Treg was unchanged in our iPD, this data alone could not exclude the possibility of a compromised suppressive function of CD4 Treg in iPD. Therefore, we also analyzed several functional markers, such as CD45RO and phospho-S6 [pS6, reflecting mTORC1 activity[51]]. The levels of both CD45RO and pS6 were decreased among CD4 Treg in iPD (Fig. 3e). Despite the reduction in those markers, the expression of FOXP3 and CTLA4, which are decisive for maintaining suppressor function[45], remained unchanged among CD4 Treg (Fig. 3f). These data indicate that it is mainly CD8 Treg cellularity that was impaired in early-to-mid stage, especially female iPD, whereas the CD4 Treg frequency and suppressive capacity were unlikely affected.

### TEMRA & the ratios between TEMRA and specific subsets as early diagnosis markers
Considering the substantial difference in CD8 TEMRA between iPD and HC, we evaluated the potential of using CD8 TEMRA as a peripheral diagnostic biomarker. We first analyzed the possible correlation between the frequency of CD8 TEMRA and various available quantitative clinical parameters, such as age, disease duration, H&Y staging scale, UPDRS-III (Unified Parkinson's Disease Rating Scale III), LEDD (Levodopa equivalent daily dose) and MOCA (Montreal Cognitive Assessment). Although no significant correlation was observed between CD8 TEMRA and most of those clinical data, the frequency of CD8 TEMRA in PD showed a significant negative correlation with disease duration from initial symptoms (Fig. 4a) or clinical diagnosis (Fig. 4b). This suggests that CD8 TEMRA populations might be more involved in the early rather than the later stage of iPD. Samples from early-to-mid stage iPD could already be well distinguished from that of HC with an area under the curve (AUC) of 0.7662, based on CD8 TEMRA frequency alone (Fig. 4c). As the frequency of CD8 TEMRA negatively correlated with disease duration, we applied another receiver operating characteristic (ROC) analysis by focusing on patients diagnosed only within 5 years. An excellent AUC value of 0.8580 [the category of AUC ranges is defined elsewhere[52]] was obtained (Fig. 4d).

Since the frequency of CD8 Treg and ILC2 also showed a difference between iPD and HC, we assessed whether those subsets could provide additional diagnostic power (Fig. 4e, f). As the ratio between CD8 TEMRA and CD8 Treg might represent the CD8 effector potential in the given individual, we also integrated the ratios and the frequency of CD8 TEMRA (Fig. 4g). Neither the frequency of CD8 Treg, the frequency of ILC2, nor the ratios between CD8 TEMRA and CD8 Treg substantially increased the potential of implementing CD8 TEMRA as a biomarker, always leaving seven HC samples mixed in the 'PD area'

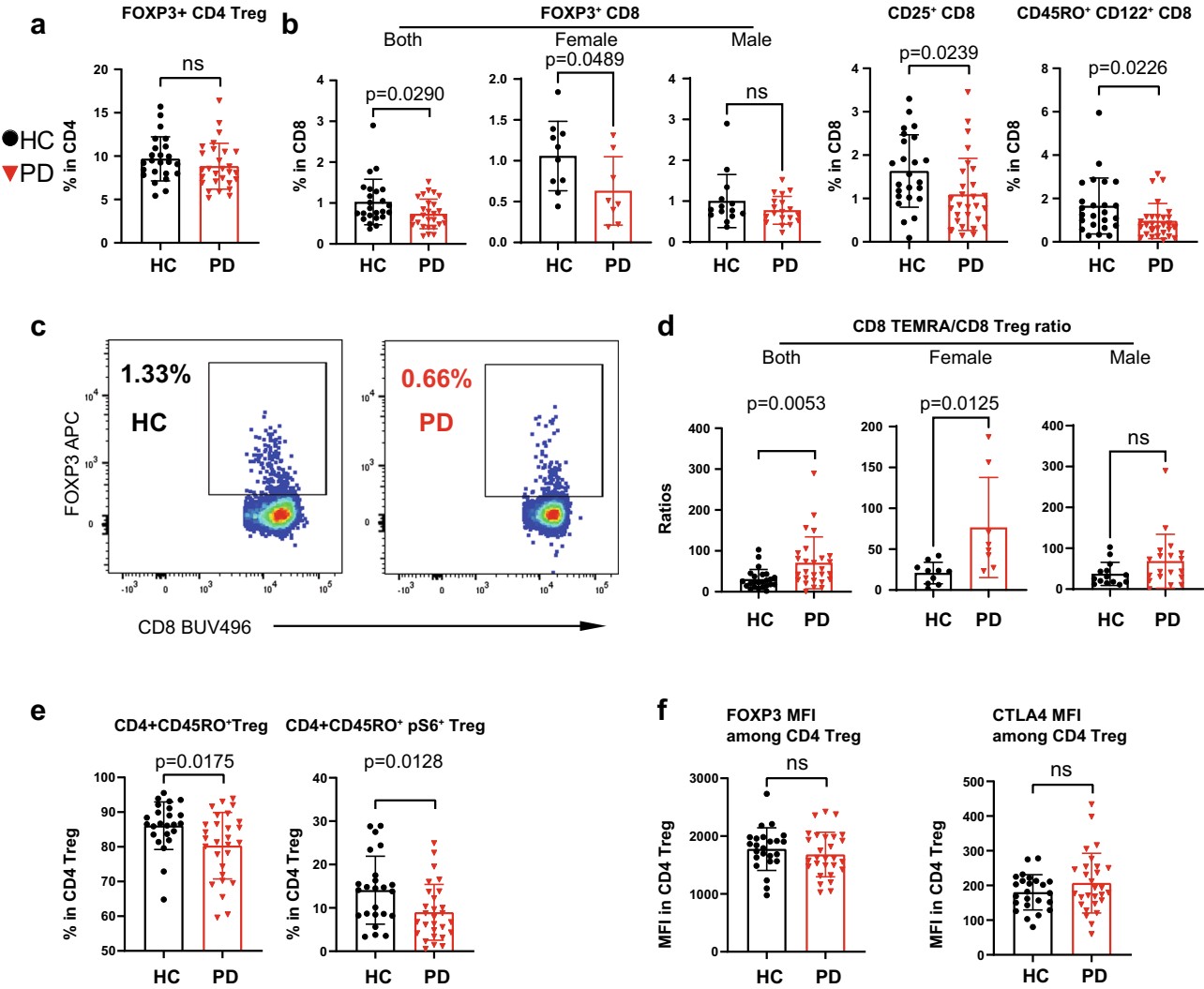

**Fig. 3 | Early-to-mid stage iPD displays a reduction in CD8 Treg but not CD4 Treg. a** Scatter dot plot showing the frequency of FOXP3⁺ cells among CD4 T cells. **b** Scatter dot plots showing the frequency of FOXP3⁺ cells for all or females or males, CD25⁺ cells and CD122⁺ cells among CD8 T cells. **c** Representative FCM plots showing the reduced frequency of FOXP3⁺ CD8 T cells (CD8 Treg) in PD vs HC. **d** Scatter dot plots showing the ratio between CD8 TEMRA (CD45RO⁻CCR7⁻CD27⁻) and CD8 Treg for all (left) or females (middle) or males (right). **e** Scatter dot plots showing the frequency of CD45RO⁺ and CD45RO⁺pS6⁺ among CD4 Treg. **f** Scatter

dot plots showing MFI of FOXP3 and CTLA4 among total CD4⁺FOXP3⁺ Treg. The results (**a, b, d–f**) were analyzed using unpaired two-tailed Student's $t$ test. Data are presented as mean ± standard deviation (s.d.). Each symbol represents the measurement from one individual (**a, b, d–f**). ns or unlabeled, not significant; all significant *P*-values are indicated. FCM flow cytometry, MFI geometric mean (geomean), Treg regulatory T cells, HC healthy controls, $n = 24$, PD patients with Parkinson's disease, $n = 28$. For **b** and **d**, female HC, $n = 10$; female PD, $n = 8$; male HC, $n = 14$; male PD, $n = 19$. Source data are provided as a Source Data file.

(Fig. 4g). A detailed analysis into the association between CD8 TEMRA to Treg ratios and the frequency of CD8 TEMRA revealed that six out of the seven 'indistinguishable' HC samples were males'. Considering this and other gender-biased results, we performed another ROC analysis with only female samples. Encouragingly, an outstanding AUC value of 0.9125 was achieved (Fig. 4h). The ROC analysis using the ratio between CD8 TEMRA and CD8 Treg also showed an excellent diagnostic potential when only females were analyzed (Fig. 4i, j). As the ratio between CD8 TEMRA and TCM was substantially increased in iPD, we also performed the ROC analysis using this ratio. Even with all the patients, an excellent AUC value of 0.8503 was obtained (Fig. 4k). With those diagnosed within five years or only females, an outstanding AUC value of 0.9470 or 0.9250 was accomplished, respectively (Fig. 4l, m). Even if only males were analyzed, an AUC value of 0.8120 was obtained (Fig. 4n). In brief, our data firmly support that CD8 TEMRA alone as well as the ratios between CD8 TEMRA and CD8 TCM or Treg are reliable, but yet unrecognized, easily-accessible cellular markers for early diagnosis of iPD, especially females.

## Validation in another subcohort & discovery of enhanced cytotoxic pathways in CD8 T cells

Given the high clinical diagnostic potential, we further validated our key findings regarding CD8 T cells in another subcohort (12 HC vs 11 iPD) of the Luxembourg Parkinson's study[22] using the FCM analysis (Supplementary Tables 2 and 6). The total CD8 T cells were unchanged among total CD3 or living lymphocyte singlets (Fig. 5a). Although all the samples of the validation subcohort were cryopreserved, the frequency of CD8 TEMRA was still clearly increased while the frequency of CD8 TCM was decreased in early-to-mid stage iPD vs HC (Fig. 5b, c). The ratio between CD8 TEMRA and TCM was still much higher in early-to-mid stage iPD than in HC (Fig. 5d). However, the ratio was generally lower in cryopreserved materials relative to fresh blood samples, possibly due to the cryopreservation effects[23,53].

To further investigate whether CD8 T cells harbor a higher cytotoxic potential in iPD, we analyzed the expression of several cytotoxicity-related markers within CD8 subsets using the FCM analysis. The frequency of GZMA−, GZMB− and perforin (PRF1)-expressing

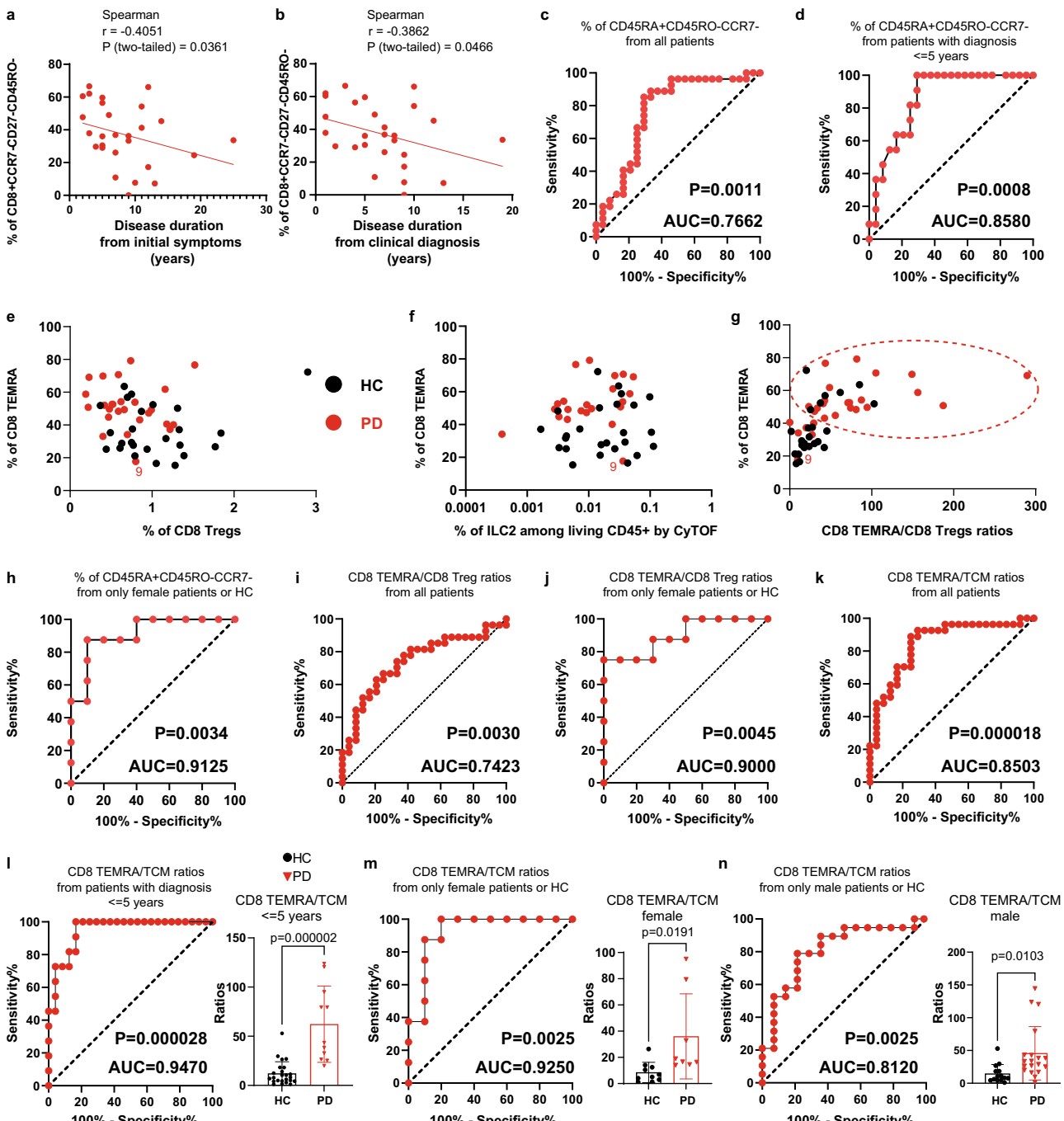

**Fig. 4 | CD8 TEMRA and the ratio between CD8 TEMRA and TCM are reliable periphery-accessible biomarkers for early-to-mid stage iPD.** Correlation between the frequency of CD8 TEMRA (CD45RO⁻CCR7⁻CD27⁻) and the disease duration from initial symptoms (**a**) or clinical diagnosis (**b**). Coefficient and *P*-value based on Spearman correlation (*n* = 28). ROC analysis based on the frequency of CD45RA⁺CD45RO⁻CCR7⁻ for all PD (*n* = 28) (**c**) or patients diagnosed within 5 years (*n* = 11) (**d**) vs all HC (*n* = 24). **e** The frequency of CD8 TEMRA (CD45RA⁺CD45RO⁻CCR7⁻) vs that of CD8 Treg. **f** The frequency of CD8 TEMRA (CD45RA⁺CD45RO⁻CCR7⁻) quantified by FCM vs the frequency of ILC2 quantified by CyTOF. **g** The frequency of CD8 (CD45RA⁺CD45RO⁻CCR7⁻) vs the ratios between CD8 TEMRA (CD45RO⁻CCR7⁻CD27⁻) and CD8 Treg. The dashed red circle highlights the PD-dominant area. **h** ROC analysis based on the frequency of CD8 TEMRA (CD45RA⁺CD45RO⁻CCR7⁻) only from female PD (*n* = 8) or HC (*n* = 10). **i, j** ROC analysis based on the ratios between the frequency of CD8 TEMRA (CD45RO⁻CCR7⁻CD27⁻) and of

CD8 Treg, for all iPD (*n* = 28) vs HC (*n* = 24) (**i**) or with only female iPD (*n* = 8) and HC (*n* = 10) (**j**). **k–n** ROC analysis based on the ratios between the frequency of TEMRA (CD45RA⁺CD45RO⁻CCR7⁻) and of TCM (CD45RO⁺CCR7⁺CD27⁺), for all iPD (*n* = 28) (**k**) or patients diagnosed within 5 years (*n* = 11) (**l**) vs all HC (*n* = 24) or only from female (**m**) iPD (*n* = 8) and HC (*n* = 10) or only from male (**n**) iPD (*n* = 19) and HC (*n* = 14). The scatter dot plot was displayed in the right panels (**l–n**). *P*-values were based on unpaired two-tailed Student's *t* test. Data are presented as mean ± standard deviation (s.d.). Each symbol represents one individual. All the PD (*n* = 28, where one female PD sample was excluded as the same patient visited twice within a short period) and HC (*n* = 24) were used unless specified. *P*-values displayed under each ROC analysis test the null hypothesis of the AUC = 0.50 with a two-tailed test. AUC area under curve, ILC2 type 2 innate lymphoid cells, ROC receiver operating characteristic, Treg regulatory T cells, TCM central memory. Source data are provided as a Source Data file.

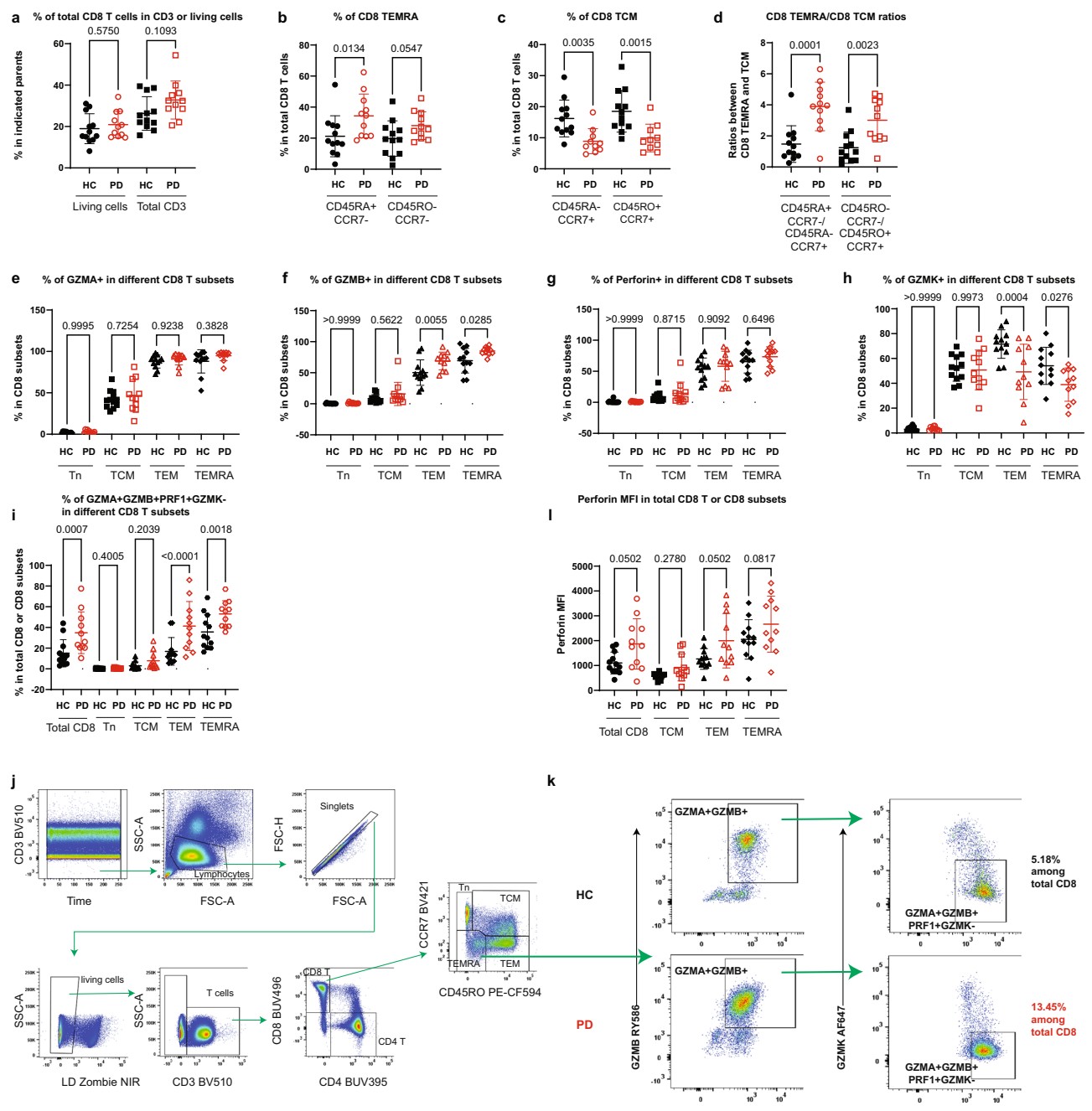

**Fig. 5 | Validation of the enhanced CD8 TEMRA portion in another subcohort and discovery of heightened co-expression of CD8 cytotoxic molecules.**
**a** Scatter dot plots showing the frequency of total CD8 T cells among CD3 or total living lymphocyte singlets using the FCM analysis of cryopreserved PBMC from another subcohort of the Luxembourg Parkinson's study (11 iPD vs 12 HC, all CMV+ except for two seronegative HC). Scatter dot plots showing the frequency of CD8 TEMRA (**b**) or TCM (**c**) among total CD8 T cells using different marker combinations. Of note, one sample was excluded from PD as it was identified as an outlier by the default setting of the ROUT method of Graphpad. **d** Scatter dot plots showing the ratios between CD8 TEMRA and CD8 TCM using different marker definitions. Scatter dot plots showing the frequency of GZMA+ (**e**), GZMB+ (**f**), perforin+/PRF1+ (**g**) or GZMK+ cells (**h**) among various CD8 T subsets. **i** Scatter dot plots showing the

frequency of GZMA+GZMB+PRF1+GZMK- cells among total CD8 T cells or CD8 subsets. **j** Gating strategy was used to identify different CD8 subsets by FCM (BD FACSymphony™ S6). **k** Representative FCM plots showing the expression of GZMA, GZMB, PRF1 and GZMK among CD8 TEMRA. **l** Scatter dot plots showing MFI of perforin among total CD8 cells or CD8 subsets. The results in **a**–**i**, **l** were analyzed using ordinary one-way ANOVA test with two-stage linear step-up procedure of Benjamini/Krieger/Yekutieli correction. q-values (FDR) were displayed. Data are presented as mean ± standard deviation (s.d.). Each symbol represents the measurement from one individual. all q-values are indicated. CMV Cytomegalovirus, FCM flow cytometry, GZMA/B/K Granzyme A/B/K, HC healthy controls, n = 12 including 5 males; PD patients with Parkinson's disease, n = 11 including 8 males. Source data are provided as a Source Data file.

cells gradually increased along the differentiation trajectory of CD8 T cells from naive (Tn) to TCM, TEM and TEMRA (Fig. 5e–g). The frequency of GZMB-expressing cells was selectively enhanced only in TEM and TEMRA of iPD vs HC. In contrast, the frequency of GZMK-

expressing cells was reduced in TEM and TEMRA (Fig. 5h). As the frequency of circulating exhausted-like GZMK+CD8+ increases during ageing[54], a reduction in GZMK expression might point towards less-exhausted and more-active 'younger' CD8 T cells in iPD vs HC. We next

quantified the combinations defined by the expression of several cytotoxic markers in CD8 T cells, as the synergistic effects might be required between relevant cytotoxic molecules to effectively perform a cytotoxic function[55]. Encouragingly, the frequency of GZMA+GZMB+PRF1+GZMK- cells was increased among total CD8 T cells, especially in TEMRA and TEM in iPD vs HC (Fig. 5i). Indeed, a largely mutually exclusive expression pattern between GZMK and other examined cytotoxic molecules (GZMA, GZMB and PRF1) was observed within CD8 TEMRA (Fig. 5j, k). Although the frequency of PRF1-expressing cells was not significantly changed in CD8 subsets, the MFI of PRF1 still showed a trend to be augmented in iPD vs HC among total CD8 T cells, TEM and TEMRA (Fig. 5l). Collectively, these cytometry data reveal that the cytotoxic function of CD8 T cells, especially the synergy among cytotoxic molecules of CD8 TEMRA and TEM, is enhanced in early-to-mid stage iPD.

To more comprehensively characterize the transcriptional program in CD8 T cells and gain mechanistic insights, we performed single-cell RNA-sequencing (scRNA-seq) based on the four sorted subsets of CD8 T cells. The reason to first sort the four subsets prior to scRNA-seq is that the decisive markers (CD45RA/CD45RO isoforms) of our target cell types are encoded by the same gene, which otherwise cannot be easily distinguished by the widely-employed shotgun sequencing methods. Considering our female-biased results in CD8 T cells, we selected only females for scRNA-seq (Supplementary Table 2). After various quality control steps, a total of 24,832 fluorescence-activated cell sorting (FACS)-isolated individual CD8 T cells from cryopreserved PBMC of iPD (*n* = 5) and HC (*n* = 4) were kept for further analysis (Fig. 6a, Supplementary Fig. 6A). Mapping the four sorted subsets with the two-dimensional uniform manifold approximation and projection (UMAP) showed a clear separation between CD8 Tn, TCM and the remaining CD8 subsets while TEM and TEMRA were partially overlapped (Fig. 6b). As expected, a higher expression of CCR7, CD27, CD28, IL7R and SELL/CD62L was observed in CD8 Tn and TCM compared with the later-differentiated ones (i.e., CD8 TEM and TEMRA) (Fig. 6c). The same expression pattern was detected for TCF7 (also known as TCF1) and TOX between terminally-differentiated exhausted CD8 T cells[56] and CD8 TEMRA compared to less-differentiated subsets (Fig. 6c). In line with our FCM results, the expression of several cytotoxic effector molecules, such as GZMA, GZMB, GZMH, GZMN, PRF1, NKG7, KLRD1, KLRG1, and especially GNLY, was enhanced in CD8 TEM and TEMRA of iPD vs HC (Fig. 6d). The portions of the cells co-expressing two or more various cytotoxic or effector molecules, such as GZMA, GZMB, PRF1 and IFNG, were also increased in iPD (Fig. 6e). This pattern mainly occurred in CD8 TEM and TEMRA without pre-stimulation (comparing Fig. 6b with Fig. 6f–i, Supplementary Fig. 6B–D). The enrichment analysis revealed that the pathways involved in graft-versus-host disease, antigen processing and presentation (e.g., CD74 and TAPBP), leukocyte trans-endothelial migration and cell adhesion (e.g., ITGB2, ITGB7, ITGAL, CD2 and CD226) and NK-mediated cytotoxicity (e.g., GZMB, PRF1, KLRD1, ITGAL, ITGB2 and PLCG2) were most altered among the upregulated genes in CD8 TEMRA of iPD vs HC (Fig. 6j–m). The gene THEMIS is a positive regulator of TCR signaling strength of peripheral CD8 T cells in response to low-affinity self-peptide MHC and cytokines[57]. THEMIS was among the most upregulated genes in CD8 TEMRA of iPD (Fig. 6j). Concurrently, several genes, critically involving in the mRNA de-adenylation-dependent degradation (enrichment *P* = 7.63E-8 among downregulated genes, Fisher's exact test, GO process), such as CNOT6L[58], ZFP36[59] and ZFP36L2, were substantially downregulated in CD8 TEMRA of iPD vs HC (Fig. 6J, Supplementary Fig. 6E). The down-regulation of this decay process potentially increases the mRNA stability in CD8 TEMRA to be better equipped for the fast-track translational steps.

Our clustering analysis within CD8 TEMRA identified two clusters (C0 and C1) (Fig. 6n). C0 expressed genes encoding proteins with effector or TCR signal activating functions [CAMK4, GZMK, COTL1[60] and THEMIS] (Fig. 6o). Although GZMK can be considered an exhaustion marker either during ageing[54] or among CCR7+ TCM[61], it is also used to identify TEM-like populations in CCR7- cells. In line with the latter, the two key exhaustion markers HAVCR2/TIM3 and CTLA4 were barely expressed in CD8 TEMRA (Supplementary Fig. 6F). The expression of the other exhaustion markers TIGIT or LAG3 was either much lower in C0 or comparable between patients and controls, respectively (Supplementary Fig. 6F). Meanwhile, PDCD1/PD-1 mRNA was undetectable in our samples. Thus, our data support C0 as a more TEM-like and non-exhausted subset. C1 was characterized by a high expression of several NK functional markers [KLRC2, KLRC3, FCGR3A/CD16, IKZF2/HELIOS and TYROBP/DAP12[62]] (Fig. 6o). The analysis of differentially-expressed genes (DEG) in iPD vs HC further demonstrated that the genes involved in NK-mediated cytotoxicity, such as GZMB, PRF1 and FCGR3A, were among the most upregulated ones in C1 (Fig. 6p). Several MHC type II molecules (e.g., HLA-DRB1, HLA-DQA1 and HLA-DQB1), known CD8 T-cell activation markers[63] as well as another NK cytotoxic effector molecule (KLRD1) and leukocyte trans-endothelial migration and adhesion molecules (ITGB2 and ITGAL) were upregulated in C0 and, to a lesser extent in C1, of CD8 TEMRA (Fig. 6p). Together, these scRNA-seq data support that the cytotoxic pathways are enhanced in CD8 TEMRA from early-to-mid stage iPD vs HC.

Similar pathways enriched for the upregulated genes in iPD vs HC were observed in CD8 TEM (Supplementary Fig. 6G–I). Our volcano plot (Supplementary Fig. 6G) showed that, AOAH (acyloxyacyl hydro-lase), a lysosomal enzyme known to detoxify LPS[64] and several established cytotoxic molecules (GZMH, NKG7 and KLRG1), were substantially upregulated in CD8 TEM of iPD vs HC. Our clustering analysis within CD8 TEM identified three clusters (C0, C1 and C2) (Supplementary Fig. 6J). Both C0 and C2 of CD8 TEM produced IFNG mRNA without pre-stimulation, while only C1 expressed the markers of CD161+/KLRB1+ tissue-homing CD8 cells, including IL23R, CCR6, SLC4A10, ZBTB16, COLQ and CEBPD[65,66] (Supplementary Fig. 6K). C0 expressed high levels of both cytotoxic molecules (GZMB, GZMH and GNLY) and the terminal-differentiation-related transcription factor ZEB2[67], indicative of a more terminally-differentiated status (Supplementary Fig. 6K). C2 showed an expression profile reminiscent of NK-like cells (including KLRC2, KLRC3, TYROBP/DAP12, LYN and IKZF2/HELIOS). Our DEG analysis shows that many cytotoxic molecules such as GZMA, GZMB, GZMH, GZMM, NKG7, GNLY and KLRD1 as well as several MHC-II genes were upregulated in at least one of the clusters of CD8 TEM in iPD vs HC (Supplementary Fig. 6L). TGFB1 and its down-stream signaling molecules, such as SMAD7, SKI and SKIL, which inhibit the expression of CD8 key effector molecules[68], were all substantially downregulated in all the clusters of CD8 TEM (Supplementary Fig. 6M). This again indicates a more-active effector status. Concomitantly, pro-survival or anti-apoptotic genes such as LMNA, GADD45B and BCL2 were downregulated, indicative of more short-lived CD8 TEM in iPD (Supplementary Fig. 6M). In total, the cytotoxic pathways in CD8 TEM were also upregulated in early-to-mid stage iPD vs HC.

## Reprogrammed CD8 Tn & TCM to favor terminal differentiation in early-to-mid stage iPD

Since the ratios between CD8 TEMRA and TCM were substantially enhanced in iPD, while total CD8 T cells were kept intact, we reasoned that the CD8 differentiation process might be affected. To unbiasedly quantify the CD8 differentiation process, we performed pseudotime trajectory analysis in our CD8 scRNA-seq datasets using the tool Slingshot[69]. Among the most significantly-differentially-expressed genes along the CD8 differentiation trajectory, the marker genes known to change over the CD8 differentiation stages, such as IL7R, LEF1 and GZMA, were confirmed by our unsupervised analysis (Fig. 7a). Encouragingly, the mean pseudotime increased from pre-sorted CD8 Tn to TCM, TEM and TEMRA (Fig. 7b). The relative fractions of cells for

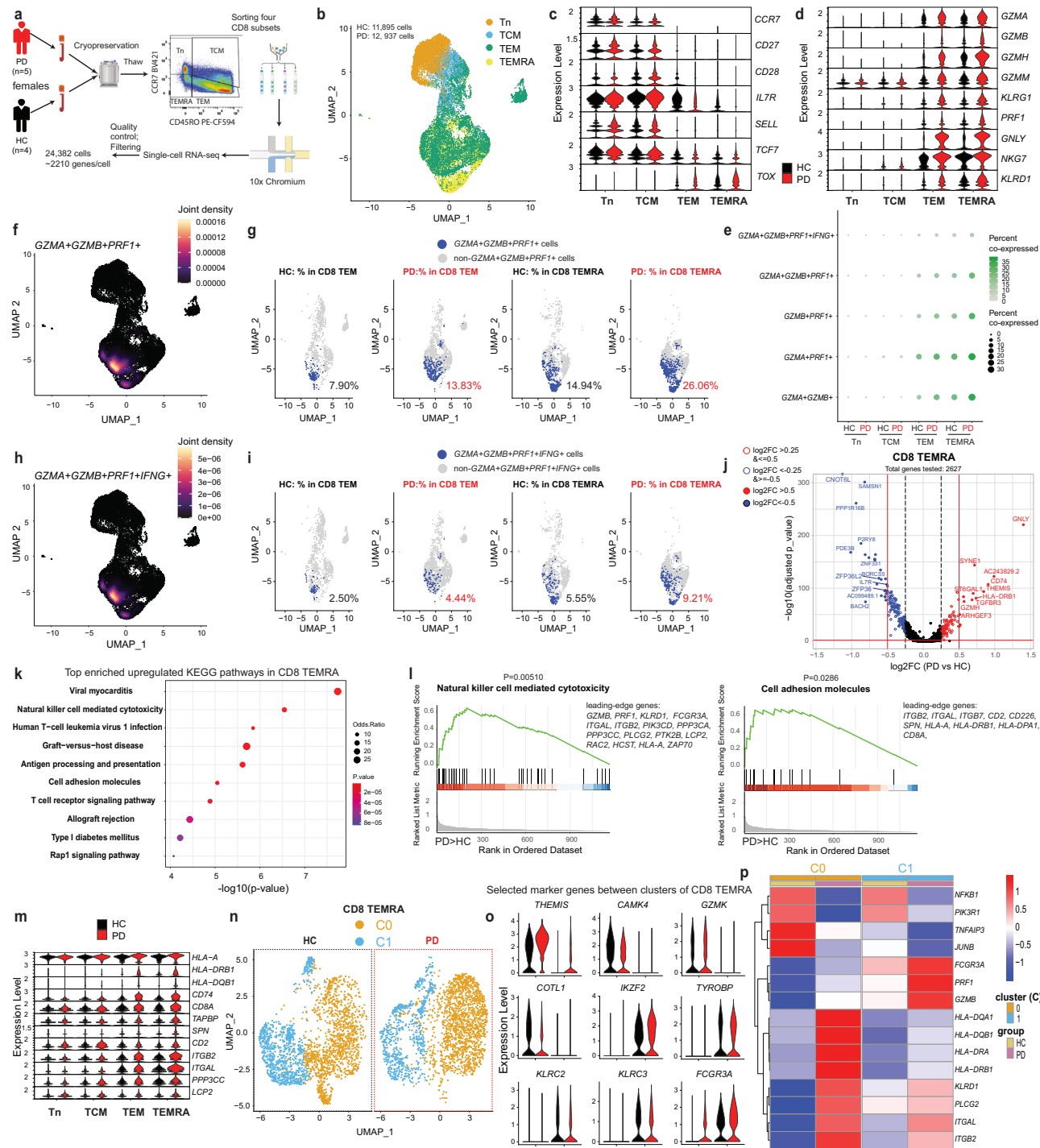

**Fig. 6 | scRNA-seq reveals enhanced cytotoxic pathways in CD8 TEMRA of early-to-mid stage iPD. a** Schematic for CD8 scRNA-seq. The other gates are in Supplementary Fig. 6a. **b** UMAP showing distribution of *n* = ~25,000 cells. Violin plots of selected genes defining CD8 naive/memory subsets (**c**), signifying cytotoxicity (**d**) and involving in top-ranked pathways enriched among upregulated genes in CD8 TEMRA (**m**). **e** Balloon plot showing the percentages of cells co-expressing the indicated markers. **f, h** UMAP showing joint density of *GZMA⁺GZMB⁺PRF1⁺* (**f**) and *GZMA⁺GZMB⁺PRF1⁺IFNG⁺* (**h**) in all CD8 cells. UMAP showing the individual *GZMA⁺GZMB⁺PRF1⁺* (**g**) or *GZMA⁺GZMB⁺PRF1⁺IFNG⁺* (**i**) cells among CD8 subsets. For visual comparability, random downsampling was employed to display the same number of cells for different conditions and subsets. **j** Volcano plots showing the expression changes in CD8 TEMRA. The selected top up- or downregulated DEGs are labeled in red or blue. Vertical dashed line indicates the log2FC value of 0.5 or −0.5, while the horizontal red line indicates −log10(0.05). **k** Top-ranked enriched KEGG pathways among upregulated DEGs in CD8 TEMRA of iPD vs HC. **l** Top-ranked

GSEA pathways in upregulated DEGs in CD8 TEMRA. The lower part showing the rank distribution of the genes involved in the indicated pathway. The list on the right showing the leading-edge genes. **n** UMAP showing the unsupervised clustering analysis for *n* = ~7200 CD8 TEMRA cells. The clustering results were split for iPD or HC cells. Cells were randomly down-sampled to the same number for iPD and HC. **o** Violin plots of selected markers distinguishing clusters. **p** Heatmap of selected top DEGs in the clusters of CD8 TEMRA. *P*-values in **j** and **p** were analyzed using two-tailed nonparametric Wilcoxon Rank Sum test adjusted based on Bonferroni correction. In **p**, only the genes with adjusted P-value ≤0.05 were considered. *P*-values in **k** and **l** were analyzed using the one-tailed Fisher's exact test and the one-tailed empirical phenotype-based permutation test, respectively. DEG differentially expressed genes, FACS fluorescence-activated cell sorting, FC fold change, *GZMA/B/K* Granzyme A/B/K, *IFNG* interferon gamma, UMAP uniform manifold approximation and projection. Panel **a** was created with BioRender.com.

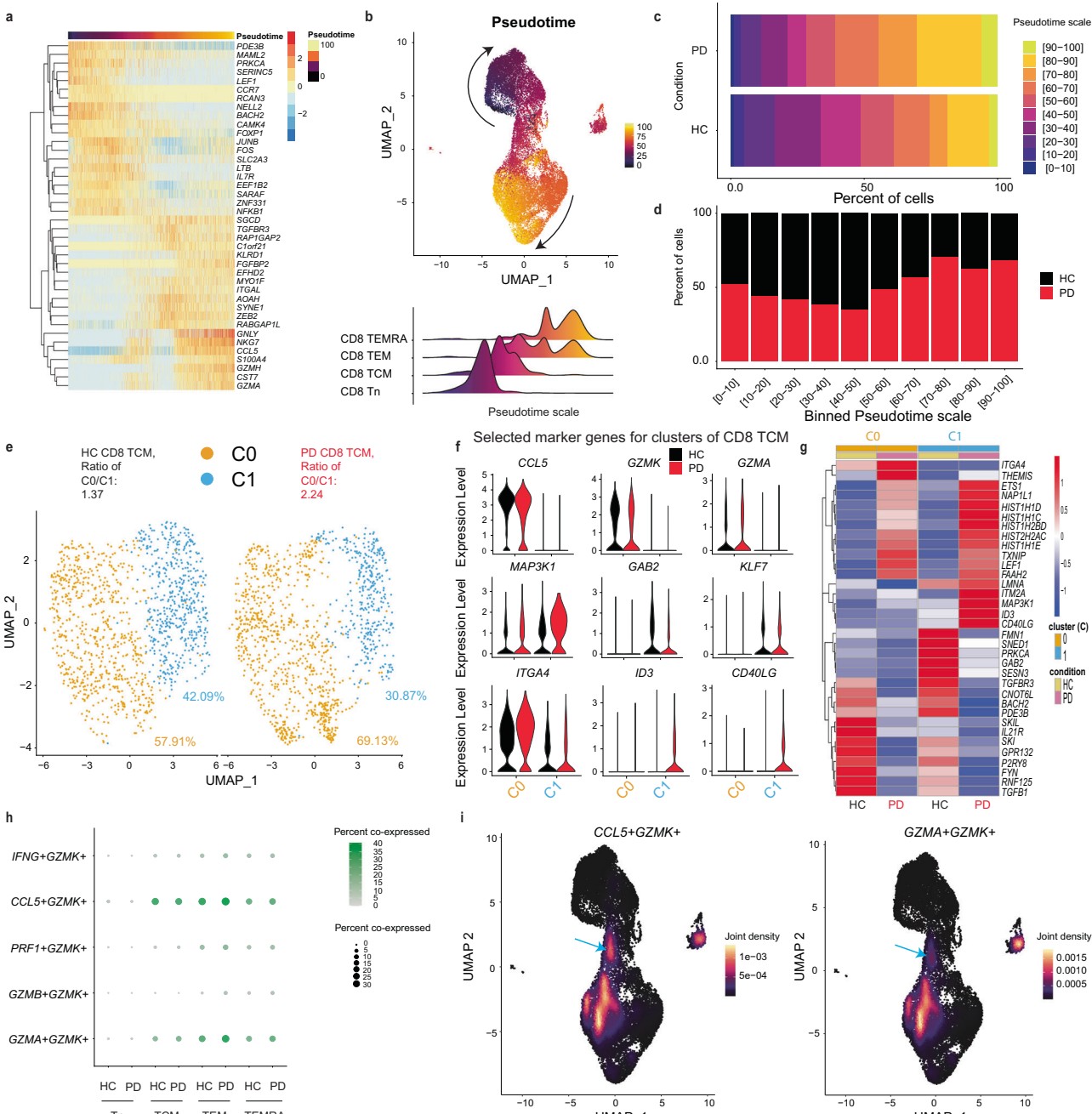

**Fig. 7 | scRNA-seq discloses the accelerated CD8 differentiation process in early-to-mid stage iPD. a** Heatmap showing the top 20 most-increased and -decreased genes along the predicted pseudotime trajectory ranked using log2FC (if *p*-value < 0.01) (*n* = ~25,000 cells). Each column represents one single cell. **b** Two-dimensional representation showing the CD8 memory T-cell differentiation trajectory using UMAP (upper). Lower, the ridge plots of pseudotime distribution among different CD8 subsets along differentiation trajectory. **c** Percentages of each binned psedutotime window along the pseudotime trajectory in iPD (*n* = ~12,000 cells) or HC (*n* = ~13,000 cells). **d** Stacked barplot showing the relative fraction of cells at the given predicted differentiation stage (during the given pseudotime window) between iPD and HC. **e** Unsupervised clustering analysis of CD8 TCM. Percentages of the two clusters and the ratios between two clusters were displayed. For comparability in visualization, cell numbers were randomly down-sampled to the same number for different conditions and subsets. **f** Violin plots of selected

marker genes distinguishing different clusters within CD8 TCM. **g** Heatmap of selected top up- or downregulated genes in CD8 TCM from iPD vs HC. **h** Balloon plot showing the percentage of cells co-expressing the indicated markers in the given subset. If a cell shows the read count equal to or higher than 1 for each of the markers in the indicated combination, it is regarded as the cell co-expressing the set of markers. **i** UMAP plot showing joint density of *CCL5* and *GZMK* (left) and of *GZMA* and *GZMK* (right) in all the individual CD8 T cells. Arrows refer to the area of TCM co-expressing the indicated markers. For **a**, the one-tailed ANOVA method was used to test whether any of the spline coefficients (for pseudotime fitting) is non-zero. In **g**, differential expression was analyzed using two-tailed nonparametric Wilcoxon Rank Sum test based on Bonferroni correction (only those with adjusted *P*-value ≤0.05 were considered). Each dot represents one single cell in (**b**, **e**, **i**). FC fold change, HC healthy controls, PD patients with Parkinson's disease, UMAP uniform manifold approximation and projection.

several mid-to-later pseudotime windows (>60) were much higher in CD8 T cells of iPD vs HC, while those in earlier windows (20-50) were lower (Fig. 7c, d). Hence, our unsupervised pseudotime analysis also demonstrates an accelerated differentiation process within CD8 T compartments of early-to-mid stage iPD.

As pseudotime analysis showed an effect already at an earlier differentiation stage, we next assessed transcriptome in CD8 TCM to characterize potential mechanisms. An unsupervised analysis within CD8 TCM identified two clusters (C0 and C1). The more-active cluster C0 was characterized by high levels of cytotoxic or proinflammatory effector molecules (e.g., *GZMA* and *CCL5*), while the resting cluster C1 was devoid of the expression of those effector molecules. The ratios between the more-active cluster C0 and the relatively-resting cluster C1 almost doubled in iPD than HC (Fig. 7e, f). The integrin *ITGA4*, known to have a higher expression in CD8 TEM compared to TCM, was highly expressed in C0 (Fig. 7f). The less-active C1 also expressed the TCR inhibitory docking adapter (*GAB2*)[70] and the transcription factor *KLF7* that is highly expressed in CD8 Tn[71] (Fig. 7f). Moreover, C1 over-expressed the antigen-specific memory T-cell marker *CD40LG* in iPD vs HC (Fig. 7f), again indicative of C1 as a quiescent TCM subset. Meanwhile, *MAP3K1/MEKK1*, the negative regulator of virus-specific CD8 T cells[72], was mainly expressed in C1, especially for iPD (Fig. 7f). In summary, our clustering analysis alone already indicates a more-active status of CD8 TCM in iPD.

To identify the potential transcriptional change in iPD vs HC, we further performed DEG analysis within CD8 TCM. First, a decrease in the fraction of cells expressing *GAB2* was observed in C1, further indicating a more-active status of CD8 TCM (Fig. 7f). Meanwhile, *THEMIS* was upregulated in both clusters of CD8 TCM of iPD (Supplementary Fig. 7A). Several critical transcription factors (*ID3*, *LEF1* and *ETS1*) that favor more the memory T-cell development[71,73,74] were also upregulated (Fig. 7f, g, Supplementary Fig. 7A). Furthermore, our DEG analysis showed that three out of five somatic linker histone H1 family genes (*HIST1H1C/H1.2*, *HIST1H1D/H1.3* and *HIST1H1E/H1.4*)[75] and several other histone genes showed a substantial upregulation in CD8 TCM of iPD (Fig. 7g). Specific subtypes of the liner histone genes (H1.2 and H1.4) have been recently shown to facilitate the differentiation of another terminally-differentiated immune cell type (i.e., neutrophils)[76]. This indicates a reprogramming of the chromatin-access status within CD8 TCM as the epigenetic landscape is known to substantially change during the CD8 memory differentiation process[77]. Although CCR7+GZMK+ cells have been identified as exhausted-like CD8 memory T-cell progenitors[61], none of the analyzed exhaustion markers (*TIGIT*, *TIM3/HAVCR2*, *LAG3* and *CTLA4*) showed an expression in *GZMK*+*CCL5*+*GZMA*+ C0 in our TCM dataset (Supplementary Fig. 7A). We further examined whether *GZMA*, *CCL5* and *GZMK* are co-expressed within the same cells of CD8 TCM. The scRNA-seq confirmed that *CCL5* and *GZMK* were co-expressed in the CD8 TCM cluster area (Fig. 7h, i). *GZMK* and *GZMA* were also co-expressed, although only in a small fraction of CD8 TCM (Fig. 7h, i). Consistent with our FCM data, the expression between *GZMK* and *GZMB* was largely mutually exclusive (Supplementary Fig. 7B). In short, our unbiased scRNA-seq from different angles demonstrate that CD8 TCM were already more ready, which might promote the terminal differentiation in early-to-mid stage iPD.

To evaluate whether the accelerated differentiation originates even earlier, we next systematically analyzed scRNA-seq data within CD8 Tn. The volcano plot shows that the early-differentiation transcription factor *LEF1*[74] was among the most highly-upregulated genes in total CD8 Tn of iPD vs HC (Supplementary Fig. 7C). The pathway enrichment analysis revealed that *RUNX1*-regulated transcription related to myeloid differentiation was ranked top (odd ratio of 84.68, *P*-value = 4.71E-4) (Supplementary Fig. 7D). More precisely, *RUNX1* and *RUNX2*, reported to favor CD8 terminal differentiation[78], were upregulated in CD8 Tn of iPD vs HC (Supplementary Fig. 7E, F).

Concurrently, *RUNX3*, known to suppress the expression of both *RUNX1* and *RUNX2* genes and terminal differentiation[79], was slightly reduced in CD8 Tn (Supplementary Fig. 7F). TCR signaling was also among the top-ranked pathways enriched for upregulated genes in CD8 Tn of iPD vs HC (Supplementary Fig. 7D). Our clustering analysis identified two clusters in CD8 Tn (Supplementary Fig. 7G). C0, the main cluster (>95% of CD8 Tn), expressed *RUNX2*, while C1 expressed the effector molecule *CCL5* and more abundantly the other two markers (e.g., *FYN* and *ITGB1/CD29*) that are highly expressed in memory CD8 T cells relative to CD8 Tn[80,81]. This indicates C1 as already a more memory-like CD8 subset (Supplementary Fig. 7H). The DEG analysis within different clusters also showed that *LEF1* and *THEMIS* were upregulated in CD8 Tn of iPD vs HC. Moreover, similar to the scenario in other CD8 subsets, several members of the linker histone H1 family genes showed a substantial increase in both clusters of CD8 Tn, while *CNOT6L* was still among the most downregulated genes (Supplementary Fig. 7H). Together, our unbiased scRNA-seq analysis indicates that early-to-mid stage iPD already initiates and reprograms the transcriptional machinery that facilitates accelerated differentiation from as early as CD8 Tn.

## Discussion

It has been postulated that PD pathology might arise in the periphery and migrate to the CNS via the vagus nerve[82,83]. More recently, Kipnis and colleagues lifted the misconception around the immune privilege in the brain[84], making the peripheral immune system a reasonable suspect contributing to the development of neurodegenerative diseases. Here we showed that early-to-mid stage iPD displayed an over-active peripheral effector immune profile, as reflected by several relevant immune cell types and soluble factors. Following our observations in the discovery subcohort, we confirmed our findings in another subcohort. Furthermore, we also observed an increased co-expression of key cytotoxic effector molecules (i.e., GZMA, GZMB and perforin) within both CD8 TERMA and TEM. This notion was further supported by our unbiased scRNA-seq analysis. At the protein level, CD8 TEMRA did not display any significant reduction in the expression of major chemokine receptors and brain homing factors in PD. In fact, our scRNA-seq analysis in patients' CD8 TEMRA and TEM even showed an enhancement of the pathway involved in the interactions between lymphocytes and endothelial cells. This could enable those T cells to cross the endothelium of the blood brain barrier and the blood-spinal cord barrier. Consistent with our cytometry data, our scRNA-seq analysis further quantitatively demonstrated accelerated CD8 differentiation in early-to-mid stage iPD. This skewed differentiation most likely originated from already-reprogrammed and more-active CD8 TCM and Tn. The "readiness" of those already transcriptionally-reprogrammed CD8 TCM and Tn might favor CD8 terminal differentiation. Without mentioning other contributing factors, the substantial upregulation of *THEMIS* in both CD8 TCM and Tn might be already sufficient to make them hyperactive to low-affinity self-peptides[57] [e.g., certain peptides or forms of α-syn[12]] and therefore to favor terminal differentiation in iPD. The elevated cytotoxic immune profile in our iPD was reflected not only by CD8 TEMRA. For instance, a reduced frequency of CD56highCD57- immature NK cells was also observed in our cohort, indicative of an accelerated differentiation within NK cells. This was further supported by the enhanced frequency of another terminally-differentiated cell type (i.e., neutrophils) and late-differentiated cells (i.e., NKT). In the meantime, we observed increased levels of specific circulating cytotoxic molecules. These data together firmly demonstrate a more-cytotoxic peripheral milieu in early-to-mid stage iPD. This might also well explain why patients with PD are better protected against some, although not all, cancers[85].

Concurrently, we also observed a reduced frequency of several regulatory or tolerance-inducing cell types. Our work provides the initial convincing evidence showing a reduced frequency of the less-

characterized immune suppressors CD8 Treg[50,86] in early-to-mid stage iPD, especially in female patients. Another work has recently shown a reduction in several regulatory cell types, including IL-10 producing CD8 cells, in PD[87]. However, they did not control for several critical confounding factors as we did here, nor did they perform gender-specific analyses. So far, our data still cannot differentiate whether the abundant CD8 TEMRA in iPD are specific against PD-relevant antigens or develop unspecifically due to the insufficient CD8 Treg control. Recently, highly expanded CD8 T cells are reported to share the same TCR clonotypes between the periphery and cerebrospinal fluid (CSF)[20]. However, as blood and CSF samples have been drawn from different individuals, more efforts are still required in this direction. The enhanced cytotoxic milieu in iPD might be analogous to a 'younger' immune system, which has been recently described by us in the patients and murine models with the deficiency of PARK7/DJ-1. Mechanistically, these similar immune dysregulations might be all stemmed via the immunometabolic processes[17,88]. Supporting this notion, mitochondrial deficiency is indeed widely reported in both idiopathic and genetic PD[89–91]. Furthermore, the frequency of the tolerance-induction subset ILC2[92] showed the strongest decrease in our early-to-mid stage iPD. Infiltration of ILC2 into CNS has been reported to improve the cognitive function in aged mice[93] and hence ILC2 reduction might also contribute to the pathogenesis of iPD. However, the relationship between peripheral and CNS-residing ILC2 in PD merits further investigation, as ILC2 are the most prevalent brain-resident ILCs in adult mice[94].

Unexpectedly, our results exhibited a certain gender-bias. While CD8 TEMRA showed an increase in both female and male early-to-mid stage iPD, although the incidence rate of PD among men is much higher[95], changes in CD8 Treg, CD8 NKT and serum GZMA were more pronounced in females. In general, women are much more susceptible to various autoimmune diseases[96–98]. Although more effort is still required to establish the autoimmune axis in iPD pathogenesis, our work already suggests CD8 TEMRA as the potential autoimmune reaction culprits, especially for female iPD. Conversely, two types of granulocytes (neutrophils and eosinophils) and ILC2 surprisingly showed more male-biased changes, which warrants more investigation. In line with this, male vs female murine neutrophils express more abundant granule genes[99].

Our data is nicely supported by other studies, but some differences are also noted. One recent scRNA-seq and TCR repertoire analysis has reported enriched terminal effector CD8 T cells in PD, although only analyzed a single-digit number of patients[20]. Furthermore, a lack of disease stage information and other relevant clinical metadata makes a direct comparison between their results and ours difficult. Similar to ours, another study[100] also observed no major changes within CD4 T cells. Nevertheless, in contrast to that scRNA-seq work and ours, that study showed a reduced frequency of CD8 CD57+ T cells and TEMRA in PD[100]. The differences between their observations and ours might be attributable to very different inclusion/exclusion criteria. In that study[100], a wider range of patients aged between 55 and 80 years old were included, while we focused on a tightly controlled group of participants (60–70 years old for iPD and HC). The age range has a major impact on immunological parameters[101,102]. Furthermore, as CMV drives immunoageing[103], we controlled for the CMV serostatus. Additionally, as peripheral immune disturbances are common in patients with cancer[104], we excluded those with cancer comorbidity, which was also unspecified in that[100] and often other PD studies.

Our work strongly suggests that CD8 TEMRA alone or, even better, the ratios between CD8 TEMRA and TCM represent valuable peripheral non-invasive biomarkers for early diagnosis, especially for female iPD. Our systems-immunology analyses suggest that CD8 TEMRA could serve as a potent periphery-accessible target to prevent the progression of iPD. Importantly, the effective intervention would only occur during a short window of opportunity, i.e., at early-to-mid

disease stage, as advised by the negative correlation between TEMRA and disease duration. Furthermore, as systematic immune profiling has rarely been performed in people at such an advanced age, our data could also serve as an immunological cell reference for various ageing-related diseases. Although technical variability among labs might impede the comparisons between our cytometric results and others, implementing standardized cytometry analysis procedures could increase comparability of inter-laboratory results[105,106]. Moreover, our data and available knowledge together encourage us to propose that upon foreign- or self-antigen activation, cross-reactive CD8 T cells could expand and differentiate into TEMRA and migrate into the CNS, where they react with neurons and other CNS-resident cells[107] and eventually kill neurons through the equipped powerful cytotoxic machinery. Meanwhile, CD8 TEMRA can secrete several cytokines, especially IFN-γ and TNF-α[24,108,109], the latter of which is enhanced in both brain and CSF of PD[110,111]. Therefore, the abundant CD8 TEMRA with uncompromised machinery to infiltrate into CNS, might be a major source of TNF-α production in CSF and brain.

## Methods
### Cohort design
We followed the ethical regulation of Luxembourg and obtained ethical approval [Luxembourg National Research Ethics Committee (CNER) approval No. 20140713-SU1; two amendment notifications on February 20, 2023 and on April 12, 2023]. Informed consent was obtained before each participant was recruited by the clinical team in Luxembourg. All study participants were recruited from the Luxembourg Parkinson's Study (https://clinicaltrials.gov/study/NCT05266872), a nation-wide, monocentric, observational longitudinal study with parallel HC. There is no financial compensation for the participants. We recruited the participants from the ongoing nation-wide Luxembourg Parkinson's study with more than 800 PD and 800 HC[22] (https://parkinson.lu/research-participation) and controlled for several major confounding factors, medications and comorbidities, known to affect the immune system, to ensure that our observations are PD-specific (Fig. 1A and Supplementary Table 1). More precisely, we excluded potential participants if they were diagnosed with any immune-associated diseases, such as autoimmune diseases, chronic inflammatory diseases, diabetes, cancer, and acute infection or if they were currently treated with immunosuppressive medication (Supplementary Table 1). Furthermore, we narrowed the patients to those at early-to-mid stage [Hoehn and Yahr (H&Y) staging scale: mean = 2.3, ranging from 1.5 to 3.0; most of them were ≤2.5 except for five participants with a scale of 3]. Since aging is the primary risk factor for PD[112] and aging dramatically affects the immune system[113], we screened for HC and iPD patients aged 60–70 years (except three genetic patients: one PD patient with two rare variants, one pathogenic homozygous variant N409S in GBA and another non-pathogenic heterozygous rare variant in PINK1 A383T, aged 48 years; one PD case with non-pathogenic heterozygous variant K13R in GBA, aged 55 years; one PD patient with the homozygous pathogenic variant L369P in PINK1, aged 45 years). After the first round of selection, 150 PD and 58 HC were further tested for their CMV serological status using biobanked serum samples. CMV has been well documented to facilitate the immunosenescence process[114]. In order to make the immunological analysis comparable at such an advanced age, we only invited HC and PD participants for fresh blood sampling if they were seropositive for anti-CMV IgG. We invited only CMV seropositive participants for both technical and epidemiological reasons. Technically speaking, we could not exclude the possibility that some CMV negative participants might become CMV seropositive during the period between the previous serum sampling time and the current fresh blood sampling time. Second, according to a large-scale CMV seropositive investigation in the neighbor areas[115], most of the elderly are CMV seropositive. Therefore, we decided to first analyze the more-representative group of participants in their sixties. Most of the

selected patients had a disease duration within 10 years from clinical diagnosis while three of them had a disease duration of 12, 13 and 19 years, respectively. As a result, a total of 28 PD and 24 HC CMV seropositive individuals consented to be included in the discovery cohort were invited for additional fresh blood sampling (see Supplementary Table 2 for details on demographic and clinical information). To account for the circadian rhythm of immune cells trafficking throughout the body[116], all blood samples of the participants were collected in the morning and processed within six hours. Sex was self-reported, supported by medical records and all the selected participants were examined by trained neurologists to remove any question in doubt. During the study design, we already considered the demographic fact that men are at a higher risk for PD than women[95] and therefore selected more men than women to participate this particular work.

As a first validation, we analyzed cryopreserved available PBMC of another subcohort from the same Luxembourg Parkinson's study. 11 iPD and 12 matched HC (Supplementary Table 2) were selected following the same inclusion/exclusion criteria as the discovery cohort. Of note, due to the availability issue of sufficient suitable samples, we included two CMV seronegative participants who otherwise met with all the other selection criteria in the HC group of the validation cohort. Excluding these two samples from the analysis did not change our conclusions (see the Peer Review file). Four HC participants were analyzed in both the initial discovery and validation analysis, but the cryopreserved samples from recent visits were used for the validation analysis, still showing certain independence even for these four samples. For scRNA-seq, considering our female-biased observations in CD8 T cells and comparability between groups of a small size, we selected only female CMV+ participants.

## Inclusion and ethics declarations

We followed the ethical regulation of Luxembourg and obtained ethical approval [Luxembourg National Research Ethics Committee (CNER) approval No. 20140713-SU1; no objection decision on two additional amendment notifications received on February 20, 2023 and on April 12, 2023]. Informed consent was obtained before the recruitment of each participant into the study by the clinical team in Luxembourg.

This single-region work was essentially performed in Luxembourg, led by Luxembourg Institute of Health in close cooperation with other Luxembourg partners (i.e., University of Luxembourg and Centre Hospitalier de Luxembourg), without involvement of other international partners. Therefore, the global research code of conduct is not applicable in this case. We also included involved clinicians and clinical scientists in the author list.

## Detection of anti-CMV IgG

To exclude a potential bias in our analysis due to a differential CMV serostatus in PD and HC, we measured the CMV serology of all the potential study participants fitting the inclusion and exclusion criteria and only selected participants who were CMV seropositive (Fig. 1). An ELISA was performed on plasma samples from previous visits that were preserved in the local biobank. We used the Human Anti-CMV IgG ELISA Kit (Abcam, ab108724) by following the manufacturer's instructions.

## Mass cytometry (CyTOF) staining and analysis

Fresh whole blood was first incubated with Human TruStain FcX™ (FcX, 422302, Biolegend) for 10 min at room temperature (RT). Surface staining was performed by transferring the blood into the Maxpar® Direct Immune Profiling Assay (MDIPA, 201325, Fluidigm, now Standardbio) tube containing a lyophilized antibody mixture. To the MDIPA-blood mixture, add four additional in-house-conjugated and two pre-conjugated antibodies (Supplementary Table 3). The four in-house-conjugated antibodies (c-Kit, KLRG1, NKP44 and CD49d) were labeled with Maxpar® X8 Antibody Labeling Kit (Fluidigm 201142A, 201159A, 201162A or 201169A, respectively). Incubation lasted for 30 min at RT. Immediately after staining was completed, Cal-lyse solution (GAS010, Thermo Fisher Scientific) was added for a 10-min incubation in the dark; then 3 mL of de-ionized water were added for another 10-min incubation in the dark. Cells were washed twice with MaxPar Cell Staining Buffer (CSB, 201068, Fluidigm) ($400 \times g$, 10 min, RT). Cells were then fixed with 1.6% of formaldehyde solution (Pierce 16% Formaldehyde, 289006, Thermo Fisher Scientific). Centrifugation conditions after fixation were $800 \times g$, for 10 min at 4 °C. As a last step, samples were incubated with Ir-Intercalator (201192A, Fluidigm), diluted (1:2000) in MaxPar Fix&Perm (201067, Fluidigm), and rested at RT for 1 h. Then, the cells were stored at −80 °C until the day of CyTOF acquisition. Prior to the acquisition, cells were washed twice with CSB and twice with Cell Acquisition Solution (CAS, 201239, Fluidigm). Cells were resuspended at 5E5 per mL in 1:10 calibration beads (EQ Four Element Calibration Beads, 201078, Fluidigm) diluted with CAS and the samples were analyzed with a HELIOS mass cytometer (Fluidigm) at a flow rate of 0.030 mL per min. Generated *.fcs files were normalized with the HELIOS acquisition software (v.7.0.8493) by using EQ beads as a standard. The frequency among living CD45+ cells well recapitulated the absolute number of cells for the given subset, as we standardized the analyzed whole blood volume (300 uL) from each participant, and no difference in the number of total living CD45+ cells was observed between PD and HC (Supplementary Table 5). Of note, due to notable staining issues, we excluded CD25, CD16 and CD127 from the relevant subset analyses as specified below. First, we did not gate and analyze Treg in our CyTOF panel (Supplementary Fig. 1). Second, for non-classic monocyte and ILC gating, we alternatively used CD38 instead of CD16 as employed elsewhere[117]. For NK subset gating, we used CD56 and CD57 to distinguish immature, mature and terminally-differentiated NK subsets[118]. Thus for different reasons, we did not use all the listed markers (Supplementary Table 3) for immune subset identification (Supplementary Fig. 1).

We mainly performed the supervised analysis based on manual gating (Supplementary Fig. 1) while we also implemented an unsupervised analysis on gated CD8 T cells. The CD8+ T cells of 50 samples (23 HC, 27 PD; one HC sample was excluded due to substantial cell loss during operation; one PD sample was excluded because a partially-wrong CyTOF staining panel was used) were extracted using FlowJo v.10 to perform the viSNE analysis on the CellEngine (https://cellengine.com/). The viSNE analysis was achieved using all the cells from each fcs file, with 1000 iterations and a perplexity of 80. The following markers were used to generate the viSNE: CD45RA/CCR7/CD27/CD57/CD38/HLA-DR. Of note, except for the results presented in Fig. 1M, all the other results were based on supervised manual analysis.

## PBMC isolation

In brief, 10-ml vacutainer K2EDTA blood collection tubes (367525, BD) was used to sample blood from each participant in the morning. Peripheral blood mononuclear cells (PBMCs) were isolated from fresh whole blood by gradient centrifugation at $1200 \times g$ for 20 min, RT using the SepMate™−50 tubes (85450, Stemcell) and Lymphoprep™ (07801, StemCell). The cells were washed three times with FCM buffer [$Ca^{2+}/Mg^{2+}$ free PBS + 2% heat-inactivated fetal bovine serum (FBS)] and counted with a CASY cell counter.

## Multi-panel multi-color flow-cytometry analysis

Similar multi-panel multi-color (up to 18) FCM analysis has already been fully established and performed by us in other human-sample-based studies[119]. With 35 phenotypic or functional markers in five panels, we were able to assess not only the proportions of different T-cell subsets, but also their functional states. To ease the comprehension, we described the major procedures here again. For each

study participant, 1 million fresh PBMCs were stained for each of the 5 staining panels. Before staining, PBMCs were incubated for 15 min at 4 °C with 50 µL Brilliant Stain Buffer (BD, 563794), containing 2.5 µL Fc blocking antibodies (BD, 564765). Fifty microliters of 2 × concentrated surface antibody mastermix diluted in Brilliant Stain Buffer were added to the cell suspension and incubated for 30 min at 4 °C (Supplementary Table 4). Washing three times with FCM buffer (300 × g, 5 min, 4 °C), the stained PBMCs were fixed for 60 min at RT using the fixation reagent of the True-Nuclear Transcription Factor Buffer Set (Biolegend, 424401). After fixation, the cells were centrifuged (400 × g, 5 min, 4 °C), resuspended in 200 µL FCM buffer and left at 4 °C overnight. The next day, PBMCs were washed once in permeabilization buffer and resuspended in permeabilization buffer containing 2.5 µL Fc blocking antibodies. After a 10-min incubation, the cells were centrifuged and resuspended in 100 µL permeabilization buffer containing the intracellular antibodies for a 30–40 min incubation at RT. Finally, the cells were washed three times in permeabilization buffer and resuspended in 100 µL of FCM buffer for the acquisition on a BD LSRFortessa™. The data were analyzed using the *FlowJo* v.10 software. Of note, with our hypothesis-free approach, we could not foresee the CD8 TEMRA results. We never used CD45RA, CCR7 and CD27 in the same panel, and this is why we had to demonstrate the CD8 TEMRA results by combining different gating strategies from different panels. Of note, in our cytometry analysis, we manually inspected the gates of each immune subset for each donor and might slightly adjust the gate positions across different donors whenever needed. Two PD samples were excluded for CD183 related results where CD183 abs was missing during staining in one panel. Furthermore, one female PD sample was excluded as the same patient visited twice within a short period.

### MSD assays to detect serological soluble factors

We used the MSD multiplex assay [U-PLEX Immuno-Oncology Group 1 (hu) Assay, K151AEL-1] to quantify the 10 selected serological soluble factors (Granzyme A, B, IFN-γ, IL-13, IL-33, IL-5, IL-8, GM-CSF, IP10 and MCP-1) following the manufacturer's recommendations similar to that in our other clinical or human-sample-based work[119]. For the MSD assay, the biobanked samples were undiluted and analyzed using the Discovery Workbench v.4.0.12 (LSR_4_0_12). For the perforin ELISA protocol, we first diluted all the serum samples 30 times, which was pre-determined by a test experiment. We used the perforin (PRF1) human ELISA kit (ab46068) purchased from Abcam to quantify serological perforin. Of note, most of the serum samples were obtained at the same time as the fresh blood taken for flow-cytometry analysis, while some of the sera were only available from the most recent visit (<one year). Some results of the participants were excluded for certain cytokines because one technical replicate was detected while another replicate was below either detection range or fit curve range.

### Flow-cytometry-based cytotoxic capacity analysis

Biobank-cryopreserved PBMC were thawed in a 37 °C water bath until the last ice crystal was visible, then rapidly transferred into a 15-mL Falcon tube containing 4 mL of heat-inactivated FBS (Gibco, 10500-064) at RT. After centrifugation (500 × g, 5 min, RT), PBMC were washed at RT with 10 mL of FCM buffer (PBS, 2%FBS; Fisher Scientific, PBS without Ca/Mg/Phenol red, 20012-027). One million cells per participant were resuspended in 100 µL of PBS and transferred into a 96-well U-bottom plate (Greiner, M9436-100Ea). Following washing twice with PBS, PBMC were incubated with 100 µL of Zombie NIR Live/dead staining reagent (Biolegend, 423106, dilution 1/200 in 1× PBS) for 30 min at RT. One hundred microliters of FCM buffer were added before centrifugation (500 × g, 4 °C, 5 min) and washed a second time. Before staining, PBMC were incubated with 50 µL of Brilliant Stain Buffer (BD, 563794), containing 2.5 µL of Human Fc Block (BD, 564765) for 15 min at 4 °C. 50 µL of surface antibody mastermix diluted in Brilliant Stain Buffer were added to the cell suspension and incubated

for 30 min at 4 °C (Supplementary Table 6). Washing twice with cold FCM buffer (500 × g, 5 min, 4 °C), the stained PBMC were then fixed for 1 h at 4 °C using 200 µL of True-Nuclear Transcription Factor Buffer Set (Biolegend, 424401). PBMC were washed twice with the permeabilization buffer and resuspended in 50 µL of permeabilization buffer containing 2.5 µL of Human FC Block. After a 10-min incubation at 4 °C, the pellet was resuspended in 100 µL of permeabilization buffer containing the intracellular antibodies and was incubated for 30 min at 4 °C (Supplementary Table 6, where we only listed all the markers included in the final analysis). Finally, the cells were washed twice in cold permeabilization buffer and resuspended in 100 µL of a freshly-prepared 4% Formaldehyde solution (Fisher Scientific, 10751395) for 30 min at 4 °C. After centrifugation, the pellet was resuspended with 200 µL of cold FCM buffer. All the stained PBMC were acquired on a BD FACSymphony™ A6 (BD Biosciences) and the FCS files were analyzed using the FlowJo v.10.8.0 software. The detailed gating strategy is illustrated in Fig. 5.

### CD8 subset sorting and scRNA-seq experimental procedures

FACS was performed using a BD FACSymphony™ S6 six-way cell sorter with a 100 µm nozzle. After counting, PBMC were stained with the protocol above used to characterize the cellular phenotype, but without proceeding to the intracellular staining steps. To preserve cell integrity to a maximum degree, the 15-mL Falcon collection tubes were extemporaneously pre-coated with a cold solution of 20% BSA in PBS for at least one hour. Before sorting, the pre-coating solution was discarded and replaced with 500 µL of 0.04% FBS in PBS at 4 °C until sorting. The sorting experiment was to separate four subsets of CD8 T cells. The gating strategy was performed as described in Fig. 6a and Supplementary Fig. 6A. The lymphocyte population was determined by their characteristic of FSC and SSC properties and then the doublets were excluded.

Samples from the same group (HC, n = 4 or iPD, n = 5) for each subset were pooled and centrifuged at 4 °C, 400 × g for 10 min. Discard supernatant, keep cells in 100 µL 0.5% BSA/PBS at 4 °C and proceed to single-cell isolation and sequencing. Cell quantification and viability were assessed using the C-Chip Disposable Hemocytometer from NanoEnTek. Cells with viability above 90% were then resuspended in PBS at a final concentration of 1000 cells/µL. The entire procedure was carried out on ice whenever possible. Single cells were processed using the Chromium Next GEM Single Cell 3′ and Library Kits v.3.1 from 10x Genomics, following the manufacturer's protocol. Libraries were purified using SPRIselect magnetic beads from Beckman Coulter, and their quality was evaluated using High Sensitivity D5000 ScreenTape Assay on an Agilent 4200 TapeStation device. Single-cell libraries with fragment sizes ranging between 450-580 bp were sequenced on Illumina NovaSeq 6000 system using the NovaSeq 6000 S1 Reagent Kit v.1.5 (200 cycles). The sequencing setup aimed to achieve 20,000 reads per cell with a paired-end configuration. The sequenced libraries were aligned to the GRCh38-2020-A human reference genome using the CellRanger (v.7.0.1).

### scRNA-seq computational preprocessing and analysis

The initial quality control involved eliminating low-quality reads. This preprocessing task was performed separately for each dataset (each subset of the group is defined as one dataset, resulting eight in total). To ensure only dataset with the highest quality was selected, we retained cells with more than 100 genes and genes that were expressed in more than 10 cells. Moreover, we excluded cells with more than 20% mitochondrial counts.

In droplet-based single-cell RNA-seq experiments, there is often background contamination resulting from a certain amount of background mRNAs. This contamination arises not only from the cells enclosed within a droplet, but also from the solution containing the cells, created through cell lysis. To correct this type of contamination,

we utilized SoupX (v.1.6.2) to estimate the cell-specific contamination fraction and adjust the expression matrix. Subsequently, we employed scDblFinder (v.1.12.0) to identify and filter out doublets. In this process, mitochondrial (number = 13), ribosomal (number = 94), and hemoglobin (number = 1) genes were removed. The integration of the four subsets was conducted using sctransform (v.2), with 3000 integration features. The integrated object comprised 34,792 cells, with a median count of genes equal to 2, 252.

We further filtered the cells based on CCR7 expression. We excluded cells that were CCR7$^+$ in the CD45RO$^+$CCR7$^-$ and CD45RO$^-$CCR7$^-$ datasets, while retaining only the CCR7+ cells in the CD45RO$^-$CCR7$^+$ and CD45RO$^+$CCR7$^+$ datasets. The remaining cell count was then reduced to 24,832, with a median count of detected genes equal to 2, 210 and a median number of detected unique molecular identifiers (UMIs) equal to 4, 805 per cell. The four CD8 subsets were then re-integrated. PCA was conducted on the integrated data, followed by a non-linear dimensionality reduction (UMAP) performed on the first 30 principal components. Density plots and joint density plots were created using Nebulosa (v.1.8.0) and scCustomize (v.1.1.1). The co-expression of multiple selected genes was visualized, demonstrating the presence of positive cells for a shared set of genes within the UMAP space. If a given cell shows the read counts equal to or higher than 1 for each of the markers in the indicated combination, the given cell is categorized as the one co-expressing the given set of markers. In the count plots, we performed random downsampling to ensure a visually-comparable sample size: 900 and 3000 cells per subset and condition for CCR7$^+$ and CCR7$^-$ subsets, respectively.

Subsequently, DEG analysis was carried out separately for each of the four CD8 subsets between HC and PD. The FindMarkers function from the Seurat package (v.4.3.0)[120] was employed. Only genes detected with a fraction value (pct) of ≥0.3 in at least one group were tested. Then we calculated two metrics for each gene, the enrichment ratio and the gene.score. The enrichment ratio was calculated for two scenarios: (i), if pct.1 > pct.2, the ratio was the fraction (pct.1) in the first group divided by that in the second group (pct.2); if pct.2 > pct.1, the ratio showing pct.2 divided by pct.1. The gene.score is a metric by multiplying the average log2 fold change and the enrichment ratio. The volcano plots of the DEGs were illustrated using ggplot2 (v.3.4.1) and ggrepel (v.0.9.2). Enrichment analysis was performed on the genes showing an adjusted $p$-value lower than 0.05 and an average log2 fold change (FC) greater than 0.25 using the enrichR library (v.3.2). Additionally, a gene-set enrichment analysis (GSEA) analysis on KEGG or Reactome pathways was conducted using clusterProfiler (v.4.6.2), where the genes were ranked based on the gene.score.

To calculate the pseudo-ordering (also known as 'pseudotime') of cells, we used slingshot (v.2.6.0) to "adjust" a one-dimensional curve intersecting with the cell subsets in the multi-dimensional expression space. We utilized low-dimensional PC coordinates for noise reduction and speed enhancement. We obtained a sequential order of cells, referred as pseudotime, based on their relative positions when projected onto the curve. Genes that were significantly different following the trajectory were found using the testPseudotime function of the TSCAN R package (v.1.36.0). The method fitted a natural spline to the expression of each gene with respect to pseudotime and to test whether any of the spline coefficients are non-zero with an ANOVA method. The top 20 up- and downregulated genes were selected based on log2 fold change (if $p$-value < 0.01).

In parallel, we performed cell clustering within each CD8 subset using the Seurat function "FindNeighbors" (dims = 1:20) and "FindCluster". As each sorted subset was already highly purified, we did not aim to generate a high number of clusters. The clustering levels were visually inspected using the functions "clustree" and UMAP. Additionally, cluster-specific marker genes were identified using the Seurat function "FindAllMarkers" for each cluster. Furthermore, to determine a meaningful cluster resolution for each sub-phenotype, additional cluster merging was performed on the subsets CD45RO+CCR7$^-$ and CD45RO$^-$CCR7+ between cluster 0 and 1 as these clusters were not further distinguishable based on the identified marker genes. To identify DEGs between the conditions HC and PD within each cluster of each subset, the function "FindAllMarkers" (log2FC.threshold = 0.5, test.use = "wilcox", min.pct = 0.3) was used. Selected DEGs of interest were visualized in heatmaps or violin plots. Of note, the preprocessing for cell clustering was slightly different from that of the pathway analyses aforementioned and the dedicated scripts were provided.

## Statistical analysis
Statistics analysis of cytometry datasets was performed in Graphpad Prism v.9.0 using an unpaired two-tailed Student's $t$ test or the Mann-Whitney nonparametric unpaired test. ROC analysis, volcano plot and PCA analysis were also performed using GraphPad Prism v.9.0. The precise test method used for the different figures is also specified in the corresponding figure legends. P-value ≤0.05 was considered significant. The error bars in the related types of figures represent the standard deviation (s.d.). For a few participants, some of the clinical information was unavailable. In that case, the related analysis did not interpolate any values to replace those empty cells in the data tables. For scRNA-seq, the precise statistical methods used for different results were provided in the sections above or the corresponding figure legends.

## Reporting summary
Further information on research design is available in the Nature Portfolio Reporting Summary linked to this article.

## Data availability
The raw scRNA-seq data generated in this study have been uploaded in NCBI GEO database under the accession number GSE237254. The raw flow-cytometry and mass cytometry (CyTOF) dataset generated in this work are available via a Zenodo repository (https://doi.org/10.5281/zenodo.8382970)[121]. The GRCh38-2020-A human reference genome used by CellRanger was downloaded from 10x Genomics website. Source data are provided with this paper. Only the relevant clinical data (i.e., disease duration) that were displayed in figures are provided in Source Data. The disaggregated sex information is also available in Source Data. Source data are provided with this paper.

## Code availability
The scripts used to analyze scRNA-seq data are available through https://doi.org/10.5281/zenodo.8398047 for the clustering analysis[122] and through https://doi.org/10.5281/zenodo.8395536 for the major scRNA-seq analyses[123].

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

## Acknowledgements

We thank all the anonymous participants of the Luxembourg Parkinson's Study for their support of our research and acknowledge the joint effort of the National Centre of Excellence in Research on Parkinson's Disease (NCER-PD) Consortium members generally contributing to the Luxembourg Parkinson's Study (the full list of consortium members are provided in Supplementary Note 1). We thank the CIEC of LIH (especially Daniela Valoura Esteves), the processing and biorepository teams of IBBL (especially Wim Ammerlann and Sabrina Saracino) and the LuxGen sequencing platform (especially Arnaud Muller and Nathalie Nicot) for their support. This study was initially supported by the Luxembourg Personalized Medicine Consortium (PMC) (CoPImmunoPD, PMC/2018/01, F.Q.H.). The study was also partially supported by Luxembourg National Research Fund (FNR) CORE program grant (CORE/14/BM/8231540/GeDES, F.Q.H.), Luxembourg Government via the CoVaLux programme (M.O.), FNR AFR-RIKEN bilateral programme (TregBAR, 11228353, F.Q.H., R.B. and M.O.) and several PRIDE programme grants (PRIDE/11012546/NEXTIMMUNE, PRIDE/10907093/CRITICS and PRIDE/14254520/i2TRON, F.Q.H., R.K., M.O.) and an individual AFR grant (PHD-2015-1/9989160, N.Z. via the group of F.Q.H.). The Luxembourg Parkinson's study is funded within NCER-PD by FNR (R.K., NCER13/BM/11264123). R.K. was further supported by an Excellence Grant in Research within the FNR PEARL programme (P13/6682797). D.K. was supported by FNR through PRIDE17/12244779/PARK-QC. We also thank Fondation Jean Think for their kind support (F.Q.H.). Some icons in several schematic figures (1a and 6a) were created with BioRender.com.

## Author contributions

C.M.C. contributed to the study design, performed the experiments, data analysis and drafted the manuscript. S.C., F.H., K.G., M.K., D.R., V.T., O.D., A.B., M.G. and N.Z. performed experiments. M.H., L.P. T.M., C.P.C.G. and R.K. participated in study design and coordinated the cohort recruitment and collection of the biological samples and clinical data. P.M. contributed to genetic PD confirmation and selection. D.K. and O.H. performed computational analysis of scRNA-seq. A.S. and A.C. contributed to data analysis and interpretation. F.B., R.B., R.K. and M.O. contributed to conceptualization and design of the project. F.Q.H. obtained the initial seed funding and conceptualized the project. F.Q.H and M.O. oversaw the whole project and revised the manuscript. All the authors approved the submitted manuscript.

## Competing interests

Pending patent application on the protection of biomarkers for Parkinson's disease (patent applicant: Luxembourg Institute of Health; inventors: F.Q.H., M.O. and R.K.; EP Patent Application No. 23203381.1 entitled "Early biomarker for Parkinson's disease"). The remaining authors of this work declare no competing interests.

## Additional information

[1]Department of Infection and Immunity, Luxembourg Institute of Health (LIH), 29 Rue Henri Koch, L-4354 Esch-sur-Alzette, Luxembourg. [2]Faculty of Science, Technology and Medicine, University of Luxembourg, 2 Av. de Université, L-4365 Esch-sur-Alzette, Luxembourg. [3]National Cytometry Platform, Luxembourg Institute of Health (LIH), 29 Rue Henri Koch, L-4354 Esch-sur-Alzette, Luxembourg. [4]Luxembourg Centre for Systems Biomedicine (LCSB), University of Luxembourg, 6 Av. du Swing, L-4367 Belvaux, Luxembourg. [5]Parkinson Research Clinic, Centre Hospitalier de Luxembourg (CHL), 4 Rue Nicolas Ernest Barblé, L-1210 Luxembourg, Luxembourg. [6]Transversal Translational Medicine, Luxembourg Institute of Health (LIH), 1A-B Rue Thomas Edison, L-1445 Strassen, Luxembourg. [7]Integrated Biobank of Luxembourg (IBBL), Luxembourg Institute of Health (LIH), 1 Rue Louis Rech, L-3555 Dudelange, Luxembourg. [8]CRBIP, Institut Pasteur, Université Paris Cité, Paris, France. [9]Department of Physics and Material Science, University of Luxembourg, 162a Av. de la Faïencerie, L-1511 Luxembourg, Luxembourg. [10]Department of Neurosciences, University California San Diego School of Medicine, 9500 Gilman Drive, La Jolla, CA 92093-0662, USA. [11]Institute of Molecular Psychiatry, University of Bonn, Venusberg-Campus 1, D-53127 Bonn, Germany. [12]Department of Dermatology and Allergy Center, Odense Research Center for Anaphylaxis (ORCA), University of Southern Denmark, Odense 5000C, Denmark. [13]Data Integration and Analysis Unit, Luxembourg Institute of Health (LIH), L-1445 Strassen, Luxembourg. [14]Present address: Institute of Microbiology, ETH Zurich, Vladimir-Prelog-Weg 4, CH-8049 Zurich, Switzerland. [15]Present address: Eligo Bioscience, 111 Av. de France, 75013 Paris, France. [16]Present address: Icahn School of Medicine at Mount Sinai, New York, NY 10029-5674, USA. ✉e-mail: Markus.ollert@lih.lu; feng.he@lih.lu

