## [Peer Review File · Nature Communications]

Early-to-mid stage idiopathic Parkinson's disease shows enhanced cytotoxicity and differentiation in CD8 T-cells in femalesREVIEWER COMMENTS

Reviewer #1 (Remarks to the Author):

In their manuscript, Capelle and colleagues provide a comprehensive analysis of human blood innate and adaptive immune cells, taken from idiopathic Parkinson's disease patients (n=28) and healthy controls (n=24, age-matched). Most of the data are based on flow cytometry and mass cytometry analyses of PBMCs and whole blood. Following these analyses, their main discovery is a higher frequency of terminally differentiated effector CD8 T cells, but also late-differentiated CD8+ NKT cells and neutrophils, along with a decrease in CD8+FOXP3+ regulatory T cells and type 2 innate lymphoid cells, in idiopathic Parkinson's disease patients, suggesting that these could be used as markers for early detection of this disease. Lastly, the authors showed a negative correlation of terminally differentiated effector CD8 T cells with disease duration, especially in female patients.

I found the manuscript to be well prepared (only minor typos), straightforward, and sufficiently detailed. The selection of the patient and control cohorts, although significantly minimizing counts, was well thought out and with good reasoning. Some of the fundamental study limitations were rightly outlined in the last section of the main text. The work may have significance to the field of neuroimmunology, Parkinson's disease and other diseases of similar traits, and some of the data may indeed be used as a source for further studies.

General comments:

1. As diversity and inclusion are important factors in science, would it be possible to elaborate more about the ethnicity of the study cohorts? it might depict a limitation to the applicability of results for certain groups. Of note, is it possible that some lifestyle choices/traits that were specific to the iPD group induced the described immune signature?
2. Could the authors comment/hypothesize in more detail about the mechanisms by which the described immune milieu might contribute to the pathogenesis of idiopathic Parkinson's disease? e.g. direct and indirect effects, according to the shown results and prior studies.
3. With regard to "our data in healthy controls alone could also serve as an immunological cell reference for various age-related diseases", what are the means by which such a comparison will overcome technical/mechanical and human variability issues? i.e. variability in staining methods/techniques, materials, signal intensity, antibody labelling, analysis/gating and others, on both cytometry methods.

Technical comments:

1. Most importantly, were all samples derived and analyzed at the same time? If not, how did the authors compensate/normalize for the staining and run-dependent changes for different days in which samples were taken? might also be relevant for analysis considerations.

2. Along the manuscript there is a varying use of cell % and MFI for the identification of marker-associated changes. What are the considerations by which MFI versus cell % were used in certain cases compared to others. On the same subject, many times MFI levels are relatively very low and might be below the level of detection or biological relevance, and what is considered "a positive staining". Although the MFI depends on staining and acquisition parameters, it would be best to consider not to have any conclusion if those are indeed such low levels (could be compared with analysis of the percentage of positive cells). Several examples for low MFIs: Figure 2G 4F (right), Figure S4B + S4E (does not change the overall conclusion, but may explain the lack of differences).

3. Due to the low resolution of the images in Figure S1, I could not clearly see the mass cytometry gating strategy.

In Figure S5F - selection of only small cells may hinder the presence of possibly larger activated lymphocytes (exclusion of further doublets could be done using a second gating with SSC-A and SSC-H).

4. What was the antibody concentration/dilution used for the antibodies in Table S3?

Reviewer #2 (Remarks to the Author):

Capelle et al. detail an interesting investigation in the immune profile of individuals with PD compared to age-matched controls. They have paid close attention to their cohort design and have accounted for many potential confounding factors (immune-altering ailments, prescription medications, etc.). On top of their stringent cohort design, Capelle et al. have embraced quality methodology in sample acquisition with the use of fresh whole blood/PBMC specimens and dual CyTOF and high-parameter flow cytometry to immunoprofile PD patient leukocytes. Their finding of an enriched CD8 TEMRA population, by both mass and flow cytometry is novel and a timely discovery in the field of neuroimmunology in PD. However, there does exist some concerning inconsistencies or excluded analyses that this reviewer would like to see rectified before potential submission. These are listed below as major and minor criticisms/questions:

Major:

-Figure 3D: T-bet/Eomes gates seem to be slightly different between HC and PD representative donors, were all gates applied uniformly? If not, why?

-Ideally, a smaller replication/validation cohort (perhaps just with FCM staining) on female PD patients and age-sex-matched healthy controls should be performed to solidify the author's claims of an increased CD8 TEMRA population in early PD (and especially in female PD individuals).

Minor:

-Seems to me that Figure 1 could either be a supplemental or somehow paired down to just add to the beginning of Figure 2 (so removing a figure and having only 4 main figures now).

-The wording of the sentence (69-70) is somewhat confusing as previous sentence describes overall

decrease in CD4 T cells, then more specifically Tregs, and lastly changes in Th1 or Th17. Makes more sense to point on that there are increases in the % Th1 and Th17 T cells (those studies cited). "Not only total CD4 T cells, but also specific CD4 subsets, such as CD4 regulatory T cells (Treg), Th1 or Th17 (Kustrimovic et al. 2018, Sommer et al. 2018), have shown changes in PD patients."

-Advice changing the phrasing of "it is burning to identify" (88)

-103: "we reasoned to have a higher..."  "we reasoned that we would have a higher"

-Figure 1: The stick-figure cartoons are overlapping in the middle, creates a messy look.

-Figure 1: Recommend some form of bullet points to help organize the lists

-Figure 2Q: suggest removing background FloJo annotation (as you are already labelling ILC2/3). Also suggest renaming CD294-CD117 annotation for clarity.

-Figure 3C: Similar to 2Q, flowjo annotation crowds the figure, flowplot could also be slightly enlarged to make space for the bolded population frequencies. Overall hard to make out gating and populations being emphasized.

-Figure 3D-H: same criticisms as Fig2Q/3C.

-Figure 4E: same criticisms with flowjo annotation.

-Overall figures and graph axis/titles use too much bolding in my opinion.

-Asterisks are not centered above some of the graphs

-Line 306, "CD4 Treg have been found..." should be Tregs

-In general, the manuscript is filled with many common grammatical mistakes, advise a critical re-reading.

Reviewer #3 (Remarks to the Author):

In this research, Capelle et. al., collected fresh whole blood and PBMCs and analyzed the cell populations using CyTOF and FCM. In PCA analysis of CyTOF results did not show any distinct immune signature in PD vs control. When further analyzing the changes in population frequencies Capelle et. al., found that although there is no difference in total CD8 T cells, there is an increase in CD8 terminally differentiated cytotoxic cells (TEMRA) and a decrease in central memory CD8 T cells (TCM). Together with an increase in CD8 NKT cells and neutrophils, the author suggests an increased cytotoxic profile in PD patients' blood, yet this was not experimentally studied here. On note, the authors found higher frequencies of CD8 T-bet+ cells, which are associated with effector functions and that CD45RO (TEMRA) CD8 cells are more proliferative and activated (shown by Ki67 and HELIOS). Furthermore, these cells were found to express higher levels of granzyme A. Reduced frequency of CD8 Treg, but not CD4 Treg, were also shown (similar to previously reported data). Next the authors aimed to find if the population alterations can be used as a "diagnostic" or for "the development of a therapeutic tool" (although the authors do not show that those cells are cytotoxic or affect PD progression).

Contribution to the field

Similar to previous studies that had suggested the involvement of T cells in the pathology of PD, Capelle et. al. identified an increase in a specific cytotoxic CD8 population in the blood of early-to-mid-stage patients compared to healthy control. These findings are of interest as little is known about the

underlying mechanism of idiopathic Parkinson's disease. However, these studies are very limited in scope and utilize only FACS/CyTOF measurements. In-depth profiling of the cells (e.g., RNAseq) could shed more light on the different "programs" activated in those of cells, or functional analysis (e.g. CD8-mediated cytotoxicity and exploring the role of TEMRA CD8 T-cells in PD models) were not performed. Moreover, it is not clear and wasn't discussed nor analyzed whether the changes in the immune cell populations in the blood can also be detected in the brain of idiopathic Parkinson's patients. The manuscript is quite descriptive, lack a mechanistic/functional investigation, and some of the conclusions in the discussion section are not based on experimental data. Therefore, I cannot accept the manuscript in its current form

Minor points

1. Line 270- "...the unaffected expression levels of the analyzed chemokine receptors and major brain homing factor among CD8 TEMRA.." – in supplementary fig S4C it looks like there is a decrease in all three chemokine receptors in PD when looking at CD8 TEMRA.
2. Figure 4I- higher granzyme A levels- is the significance attributed to the outlier patient? Are there any other effector molecules to be tested?
3. Figure S5k- no significant changes in cytokines levels in the blood. If these effector cells enter the CNS, what is the changes in cytokine levels in the CSF?
4. Didn't understand how the CMV criteria was implemented in the study. Seems that maybe CMV+ patients should be excluded.
5. None-specific definitions such as "have shown changes" (line 70) should be avoided.
6. Matheoud et al. 2019 do not demonstrate the "cytotoxic CD8 T-cell response against mitochondrial antigens caused PD-like motor symptoms" (lines 79–80) but merely that Gram-negative bacteria in Pink1-/- mice engages mitochondrial antigen presentation and the establishment of cytotoxic mitochondria-specific CD8+ T cells in the periphery and in the brain.
7. Fig 2C – it is not clear how the cell identifies were defined.
8. Subject criteria- patients were characterized with early-to-mid-stage up to 10 years from diagnosis. 1 patient grade 3 was included and three patients diagnosed over than 10 years. Such patients do not follow the criteria and might alter reported results.
9. Supplementary fig S4- different abbreviations are used for chemokine receptors in the figure and text which is confusing.
10. Analyzing only the female population increased AUC score in ROC analysis. Will separation of CD8 TEMRA and CD8 Treg or using TEMRA/Treg ratio only on the female population will produce a better diagnostic criteria?
11. Looking at granzymes levels and other effector characteristics will be significant when analyzing only females?
12. Line 372- written TEMTA instead of TEMRA.
13. Discussion- CD45RA is not expressed solely by CD8 TEMRA (on naïve CD4 for example). Is there a way to "easily target" CD8 TEMRA as stated in the discussion? Or is it needed to be further investigated, therefore we still do not know if it really is an "easy" target.

REVIEWER COMMENTS

Reviewer #1 (Remarks to the Author):

In their manuscript, Capelle and colleagues provide a comprehensive analysis of human blood innate and adaptive immune cells, taken from idiopathic Parkinson's disease patients (n=28) and healthy controls (n=24, age-matched). Most of the data are based on flow cytometry and mass cytometry analyses of PBMCs and whole blood. Following these analyses, their main discovery is a higher frequency of terminally differentiated effector CD8 T cells, but also late-differentiated CD8+ NKT cells and neutrophils, along with a decrease in CD8+FOXP3+ regulatory T cells and type 2 innate lymphoid cells, in idiopathic Parkinson's disease patients, suggesting that these could be used as markers for early detection of this disease. Lastly, the authors showed a negative correlation of terminally differentiated effector CD8 T cells with disease duration, especially in female patients. I found the manuscript to be well prepared (only minor typos), straightforward, and sufficiently detailed. The selection of the patient and control cohorts, although significantly minimizing counts, was well thought out and with good reasoning. Some of the fundamental study limitations were rightly outlined in the last section of the main text. The work may have significance to the field of neuroimmunology, Parkinson's disease and other diseases of similar traits, and some of the data may indeed be used as a source for further studies.

Reply: Thanks for the general appreciation for the cohort participant selection, the study design and the significance to the field.

General comments:

1. As diversity and inclusion are important factors in science, would it be possible to elaborate more about the ethnicity of the study cohorts? it might depict a limitation to the applicability of results for certain groups. Of note, is it possible that some lifestyle choices/traits that were specific to the iPD group induced the described immune signature?

Reply: Thanks for pointing out this important aspect. Although we could speculate that most of our Parkinson patients and healthy controls are Caucasian (as this cohort is located in Luxembourg, part of the central European area), we did not officially collect this type of information. We also did not consider to collect lifestyle information in our cohort, which had not yet become a major topic when we built up the cohort almost 10 years ago.

2. Could the authors comment/hypothesize in more detail about the mechanisms by which the described immune milieu might contribute to the pathogenesis of idiopathic Parkinson's disease? e.g. direct and indirect effects, according to the shown results and prior studies.

Reply: Thanks for providing this valuable suggestion. Although avoiding far-reaching discussions, we now still hypothesized and discussed the possible contributions of CD8 TEMRA to the pathogenesis of iPD in line 648-661, page 19. Furthermore, it is worth to highlight that during the revision, we have also performed additional experiments demonstrating that those CD8 TEMRA have high cytotoxic capacity using both cytometry analysis and single-cell RNA-seq methods. These additional experiments have further helped us to tailor our discussions.

To directly answer this comment of the reviewer, we also briefly discussed the key idea here. Following Braak's hypothesis (<https://pubmed.ncbi.nlm.nih.gov/28243222/>), PD might start from the gut (<https://pubmed.ncbi.nlm.nih.gov/33649989/>) and the aggregation of the Lewis bodies would propagate through the vagus nerve to the brain. The immune system, and based on our findings regarding CD8 T cells, could get sensitized to the a-synuclein aggregates (or other possible foreign- or self-antigens) in the periphery, favour the propagation through the mounted immune response (local inflammation and cytotoxicity) and finally reach the brain.

If the original sensitization against a-synuclein happens in the periphery (either directly or through molecular mimicry of an unrelated antigen, possibly of microbiome origin), the immune cells (or following our data, the CD8 T cells) could initiate the attack against a-synuclein in the periphery and cause its propagation to the brain. To supporting this idea, even in the initial version, we already cited the relevant paper (<https://academic.oup.com/brain/article/143/12/3717/5942715>) analysing post-mortem brain tissues, where those authors wrote "Overall, our results suggest that CD8 T cells may be relevant in both the initiation and the progression of disease as they precede neuronal death and synucleinopathy."

3. With regard to "our data in healthy controls alone could also serve as an immunological cell reference for various age-related diseases", what are the means by which such a comparison will overcome technical/mechanical and human variability issues? i.e. variability in staining methods/techniques, materials, signal intensity, antibody labelling, analysis/gating and others, on both cytometry methods.

Reply: Thanks for raising this important aspect. We fully agree with the reviewer that the variability at different layers might affect the comparability of other results with our immune-cell datasets. We do agree with the reviewer that if one uses very different staining protocols, antibody concentrations and gating strategies, it might be very challenging to compare their results with ours. However, with the implementation of more and more standardized

operational protocols as discussed in various papers (<https://onlinelibrary.wiley.com/doi/full/10.1002/cyto.a.23901>; <https://www.sciencedirect.com/science/article/pii/S0022175917302053>), inter-laboratory cross-platform comparability of cytometry results is increasing. Furthermore, for mass cytometry (CyTOF) analysis, we actually utilized standardized, integrated commercialized staining assays (https://store.standardbio.com/ccrz__ProductDetails?sku=201334&cclcl=en_US), which has further enhanced the potential using our immune-cell datasets of HC as reference for tested individuals in their sixties. We would like to point out that according to the experience of ours and others, the percentage results (although MFI might be still tricky) are quite comparable between labs, even if there exist some variabilities in terms of experimental settings and analyses strategies. We have now additionally discussed this aspect **in line 745-751, page 21-22**.

Technical comments:

1. Most importantly, were all samples derived and analyzed at the same time? If not, how did the authors compensate/normalize for the staining and run-dependent changes for different days in which samples were taken? might also be relevant for analysis considerations.

Reply: Thanks for raising this critical technical point. At least for fresh samples in the initial discovery (we only used available cryopreserved PBMC samples for the suggested validation experiments during revision), all the blood samples of the recruited participants were collected in the morning between 8:00-11:00 AM, considering the diurnal changes (also known as circadian rhythm) that occurs in our immune system (<https://pubmed.ncbi.nlm.nih.gov/23391992/>). However, for fresh sample analysis, participants visited our clinics at different days. Therefore, we were unable to perform the staining and acquisition of all the fresh samples together. For the measures taken to circumvent this aspect, please see our responses below.

Our cytometry analysis was performed at the National Cytometry Platform, where dedicated skilled people take care of daily quality control, which might be not possible for many other institutions. For flow cytometry analysis, BD™ Cytometer Setup and Tracking application (CS&T) with BD FACSDiva™ software were used daily to check for instrument performance. Voltage setting relative to the panels used in the manuscript is linked to everyday performance using the Application Setting™ workflow. This ensures that voltage settings are adapted every day to minimize changes in the instrument's performance over time.

For mass cytometry (CyTOF), the instrument's performance was monitored daily with the CyTOF software using a Tuning solution (201072, Fluidigm). Samples were mixed with EQ Four Element Calibration Beads (201078, Fluidigm) as described in the Materials & Methods. EQ Four Element Calibration Beads were used to normalize signal variation occurring in the instrument over time.

Considering all of the above, we are confident that we minimized all possible time-related variations in our cytometry data. For the suggested validation experiments in the revision, we indeed measured the selected cryopreserved PBMCs together. Thus, the results of the validation experiments are not much suffering from the indicated potential issues of technical variability.

2. Along the manuscript there is a varying use of cell % and MFI for the identification of marker-associated changes. What are the considerations by which MFI versus cell % were used in certain cases compared to others. On the same subject, many times MFI levels are relatively very low and might be below the level of detection or biological relevance, and what is considered "a positive staining". Although the MFI depends on staining and acquisition parameters, it would be best to consider not to have any conclusion if those are indeed such low levels (could be compared with analysis of the percentage of positive cells). Several examples for low MFIs: Figure 2G 4F (right), Figure S4B + S4E (does not change the overall conclusion, but may explain the lack of differences).

Reply: Thanks for raising this critical technical aspect. We would like to point out that: "low-level" MFI is always somehow relative. If you increase the voltage, even the negative cell population could also have a high MFI in flow cytometry (or MSI, median signal intensity for CyTOF). Furthermore, the staining for all the markers shown in the manuscript has been checked for specificity and the capacity to distinguish positive and negative cell populations. In mass cytometry (CyTOF), the internal reference method was used whereas in flow cytometry both FMO and internal reference were used.

Last but not least, to address the concern of the reviewer on particular figures, we performed additional analysis to show the percentages of positive cells for the given figures. For **Figure 1G** (corresponding to the previous Figure 2G; in the revision, following reviewers' comments, we now removed Figure 1), we now added **Fig. S2J** to show the percentages of CD57+ cells, which still showed similar results as that shown in the new **Figure 1G**. We also now added representative cytometry plots to show the expression of CD57 among CD8 TEMRA in **Fig. S2J**. For previous Figure 4F (Corresponding to new **Figure 3F** of the revised manuscript), CTLA4 is known to be quite highly expressed in Treg and the displayed "relatively low" CTLA4 MFI is simply due to a low voltage used for that channel. In any case, there is no significant difference in CTLA4 MFI among CD4 Treg between PD and HC. As for % of CTLA4+ cells

among CD4 Treg, there was also no clear significant difference (even showing a trend to be increased in PD vs HC, see the plot below). We did not show this figure in the manuscript because these CTLA4+ % results, in our opinion, will not provide an added value to the non-significant results of CTLA4 MFI.

For the MFI results in **Fig. S4B**, the percentages of those chemokine-receptor-expressing cells were actually already shown in **Fig. S4A**. For **Fig. S4E**, considering the critical functional relevance of CD49d, we performed additional analysis and showed % of CD49d+ cells among various CD8 subsets (in the right panel of **Fig S4E**, although there was still no difference between PD and HC). In new **Fig. S4F**, we also showed the representative cytometry plots and histogram overlay between different CD8 subsets to indicate the confidence of our results.

3. Due to the low resolution of the images in Figure S1, I could not clearly see the mass cytometry gating strategy.

In Figure S5F - selection of only small cells may hinder the presence of possibly larger activated lymphocytes (exclusion of further doublets could be done using a second gating with SSC-A and SSC-H).

Reply: Thanks for pointing out this critical point. Indeed, we should have provided a high-resolution gating strategy Figure in the original submission. During the revision, we now completely re-arranged the layout of the gating strategy plots in **Fig. S1**.

For **Fig. S5F**, we have now carefully checked and discussed our gating strategy with our cytometry platform. As shown in the Figure below, when we performed backgating, it is clear that our gating strategy already included all the cells of interests, such as different CD8 T-cell subsets and specific CD4 subsets.

We have also particularly checked a second gate with SSC-A and SSC-H to see the possibility to further exclude doublets. As the Figure shown below, the second gating with

SSC-A and SSC-H will not help further to remove additional doublets, at least in our dataset. Therefore, we believe our current gating strategy presented in the manuscript is appropriate for our purpose.

4. What was the antibody concentration/dilution used for the antibodies in Table S3?

Reply: Most of the Abs we used for CyTOF analysis is part of the Maxpar® Direct™ Immune Profiling Assay (MDIPA) kit (<https://www.standardbio.com/area-of-interest/immune-profiling/maxpar-direct-immune-profiling-system>). Therefore, the optimized concentration/dilution is, part of their protected commercial information, which is unfortunately not publically available. But in the footnote of Table S3 (line 1858-1862, page 53), we now additionally provided the concentration information of the in-house conjugated Abs that we stained together with the MDIPA kit.

Reviewer #2 (Remarks to the Author):

Capelle et al. detail an interesting investigation in the immune profile of individuals with PD compared to age-matched controls. They have paid close attention to their cohort design and have accounted for many potential confounding factors (immune-altering ailments, prescription medications, etc.). On top of their stringent cohort design, Capelle et al. have embraced quality methodology in sample acquisition with the use of fresh whole blood/PBMC specimens and dual CyTOF and high-parameter flow cytometry to immunoprofile PD patient leukocytes. Their finding of an enriched CD8 TEMRA population, by both mass and flow cytometry is novel and a timely discovery in the field of neuroimmunology in PD. However, there does exist some concerning inconsistencies or excluded analyses that this reviewer would like to see rectified before potential submission. These are listed below as major and minor criticisms/questions:

Reply: Thanks for general appreciation on our cohort design, participant selection and experimental methodology control using fresh samples. We also acknowledge the reviewer for pointing out the novelty of and the significance of our timely discoveries.

Major:

-Figure 3D: T-bet/Eomes gates seem to be slightly different between HC and PD representative donors, were all gates applied uniformly? If not, why?

Reply: Thanks for notifying this important aspect. For murine cytometry analysis, the gating can more or less stay exactly the same for different samples, but in humans the profile of the immune cell populations is very different from individual to individual and the gating cannot stay unchanged in our opinion.

Whether identical gates should be applied across human donors is a long lasting discussion (<https://onlinelibrary.wiley.com/doi/full/10.1002/cyto.a.22319>). Biological variability linked to disease, gender, age, circadian rhythm or genetic background is one of the fundamental problems when drawing gates manually, since not all the variables are under control (<https://bmcbioinformatics.biomedcentral.com/articles/10.1186/s12859-020-03795-w>).

Solutions may vary from drawing wider gates and/or following the positive population keeping in account the FMO controls. Adjusting gates among different human donors following the application of a common template is therefore a common practice (https://link.springer.com/protocol/10.1007/978-1-4939-9650-6_5). To be more transparent, we now stated that we performed donor-specific adjustments in our analysis in the **Methods** (line 889-891, page 26).

-Ideally, a smaller replication/validation cohort (perhaps just with FCM staining) on female PD patients and age-sex-matched healthy controls should be performed to solidify the author's claims of an increased CD8 TEMRA population in early PD (and especially in female PD individuals).

Reply: Thanks for making this constructive suggestion. But it is not feasible for the given time frame to perform a similar analysis in a completely independent external cohort. As we are not aware of any existing cohort that has these types of cryopreserved PBMC samples available, so it would be a prospective approach with adapted ethics at partner cohorts in another country, reconsenting, prospective sampling and additional experiments, which might take at least 1.5 year in our neighbouring countries. Alternatively, we identified up to 11 iPD vs 12 age- and gender-matched HC from another subcohort of the existing Luxembourg Parkinson's study following the same stringent inclusion/exclusion criteria as the initial discovery analysis. Additionally, we obtained samples from 5 female PD vs 4 female HC for single-cell RNA-seq (scRNA-seq) analysis. As discussed in the manuscript, we only used female samples for scRNA-seq due to our female-biased observations. Moreover, in the scRNA-seq, mixing a tiny number of male and female samples together would have only compromised the analysis power. We have to highlight that performing those additional experiments is not trivial, as we

had to go through a couple of rounds of time-consuming ethical approvals, together with the experimental and computational analysis efforts, which took us around one year to finalize this validation experiment and scRNA-seq analyses.

Due to the lack of available samples from suitable female patients (partially because of the higher risk of PD for men than women, refer to the paper <https://jnnp.bmj.com/content/75/4/637>), 8 out of 11 iPD are male and all the PBMC samples were cryopreserved at our local biobank. Five out of selected 12 HC are male. Despite of this aspect, we were still able to validate our major observations that CD8 TEMRA and the ratios between CD8 TEMRA and TCM are enhanced in early-to-mid iPD vs HC (See new **Figure 5B-D**). We described the results in line 401-412, page 12. This is not really unexpected as shown in **Figure 2C** and **Fig. S3D**, at least for CD8 TEMRA and TCM, the difference between males and females was not so striking for fresh blood samples (although the results of females were indeed much more significant than male data, and female-biased results were observed for other subsets and in the ROC analysis). Furthermore, from a statistical point of view (by Chi-square test), there is no significant difference in participant gender distribution between iPD and HC in our validation analysis. To be transparent, we would like to point out the following fact. Due to the availability issue of sufficient suitable samples, we included two CMV seronegative participants who otherwise met with all the other selection criteria in the HC group of the validation cohort. Although excluding these two CMV samples from the analysis did not change our conclusions (see the Figure below), we included them in the manuscript aiming to slightly increase the statistical power. For comparison, one need to compare the new **Figure 5B-D and 5I** including the two CMV seronegative samples in HC, with the Figure below without the two samples. We also clarified this point in the *Cohort Design* of the **Methods** of the revised manuscript (line 808-817, page 23) and in the Reporting Summary.

After excluding the two HC with CMV seronegativity

For the scRNA-seq analysis, we had the privilege to access only female iPD and HC samples. The unsupervised scRNA-seq analysis clearly shows not only enhanced cytotoxicity within CD8 T cells (especially, CD8 TEMRA and TEM, **new Figure 6 and Fig. S6**), but also an accelerated CD8 differentiation process, as well as already more-active transcriptionally-reprogrammed CD8 TCM and naïve cells in early-to-mid iPD vs HC (**new Figure 7 and Fig. S7**).

In short, we were happy to be able to validate our essential observations using an independent sub-cohort of the Luxembourg Parkinson's Study.

Minor:

-Seems to me that Figure 1 could either be a supplemental or somehow paired down to just add to the beginning of Figure 2 (so removing a figure and having only 4 main figures now).

Reply: Thanks for this critical suggestion. We now decided to remove the previous Figure 1 and put the major inclusion/exclusion criteria directly in the revised **Figure 1A** (corresponding to the previous Figure 2A). One could also visit **Table S1** for the detailed inclusion and exclusion criteria used in this work.

-The wording of the sentence (69-70) is somewhat confusing as previous sentence describes overall decrease in CD4 T cells, then more specifically Tregs, and lastly changes in Th1 or Th17. Makes more sense to point on that there are increases in the % Th1 and Th17 T cells (those studies cited). “Not only total CD4 T cells, but also specific CD4 subsets, such as CD4 regulatory T cells (Treg), Th1 or Th17 (Kustrimovic et al. 2018, Sommer et al. 2018), have shown changes in PD patients.”

Reply: Thanks for identifying this point. We have now revised this sentence to make the message more specific. In fact, the two cited papers described opposite observations. We now changed the sentence (line 79-81, page 3) to “Not only total CD4 T cells, but also specific CD4 subsets, such as CD4 regulatory T cells (Treg) and Th17 have shown a reduction (Kustrimovic et al. 2018), although Th17 have been observed enhanced in PD patients by another study (Sommer et al. 2018).”

-Advice changing the phrasing of “it is burning to identify” (88)

Reply: Thanks for pointing out this. We now changed to “It is important to identify...”.

-103: “we reasoned to have a higher...”  “we reasoned that we would have a higher”

Reply: We now revised as suggested.

-Figure 1: The stick-figure cartoons are overlapping in the middle, creates a messy look.

Reply: Considering this and the comment below as well as another comment from another reviewer, we decided to remove the previous Figure 1. To further compact the already-long manuscript and directly come to the point, we now added the major inclusion and exclusion criteria directly in the new **Figure 1A** (corresponding to Figure 2 in the original submission).

-Figure 1: Recommend some form of bullet points to help organize the lists

Reply: Please refer to our reply above.

-Figure 2Q: suggest removing background FloJo annotation (as you are already labelling ILC2/3). Also suggest renaming CD294-CD117 annotation for clarity.

Reply: Thanks for making this important suggestion. As indicated above, the previous Figure 2Q corresponds to the new **Figure 1Q**. We now removed original background Flowjo annotations. We also renamed CD294 and CD117 annotations. Thanks to this suggestion, these figures all look much more clean.

-Figure 3C: Similar to 2Q, flowjo annotation crowds the figure, flowplot could also be slightly enlarged to make space for the bolded population frequencies. Overall hard to make out gating and populations being emphasized.

Reply: We now removed original background *Flowjo* annotations. Thanks to your suggestion, now the percentage numbers in the emphasized subsets should be clear in the plots. Furthermore, we also added additional dashed rectangles to highlight the gates of interests. Of note, due to the removal of the previous participant selection-related **Figure 1**. The previous Figure 3C is now called **Figure 2C**.

-Figure 3D-H: same criticisms as Fig2Q/3C.

Reply: We now did the suggested correction in the updated new **Figure 2E-I** (corresponding to Figure 3 in the original submission).

-Figure 4E: same criticisms with flowjo annotation.

Reply: We now did the suggested correction in the updated new **Figure 3C** (corresponding to Figure 4 in the original submission).

-Overall figures and graph axis/titles use too much bolding in my opinion.

Reply: Thanks for providing this opinion. According to our experience in the peer-reviewing process of our recently-published works, what we were often criticized by our peers were the displays of too small fonts in the graph axes and titles. This is why we particularly bolded and enlarged the titles/axes within Figures in this work to increase the readability. We believe this formatting aspect can be easily fixed by the typeset editors, if accepted and needed.

-Asterisks are not centered above some of the graphs

Reply: Thanks for your careful reading and checking. We now carefully checked each subpanels and tried to put the asterisks in the center of the pairs of compared groups.

-Line 306, "CD4 Treg have been found..." should be Tregs

Reply: Thanks for notifying this aspect. In fact, we particularly used the term "CD4 Treg" to differ from CD8 FOXP3+ cells, which we named as "CD8 Treg". If there had been no interesting observations about CD8 Treg, we would have directly followed the suggestion of the reviewer.

-In general, the manuscript is filled with many common grammatical mistakes, advise a critical re-reading.

Reply: We are sorry for having some common grammatical mistakes. During the revision, several senior authors have paid attention to this aspect and carefully checked through the manuscript several times. We hope the number of common grammatical mistakes have been minimized.

Reviewer #3 (Remarks to the Author):

In this research, Capelle et. al., collected fresh whole blood and PBMCs and analyzed the cell populations using CyTOF and FCM. In PCA analysis of CyTOF results did not show any distinct immune signature in PD vs control. When further analyzing the changes in population frequencies Capelle et. al., found that although there is no difference in total CD8 T cells, there is an increase in CD8 terminally differentiated cytotoxic cells (TEMRA) and a decrease in central memory CD8 T cells (TCM). Together with an increase in CD8 NKT cells and neutrophils, the author suggests an increased cytotoxic profile in PD patients' blood, yet this was not experimentally studied here. On note, the authors found higher frequencies of CD8 T-bet+ cells, which are associated with effector functions and that CD45RO (TEMRA) CD8 cells are more proliferative and activated (shown by Ki67 and HELIOS). Furthermore, these cells were found to express higher levels of granzyme A. Reduced frequency of CD8 Treg, but not CD4 Treg, were also shown (similar to previously reported data. Next the authors aimed to find if the population alterations can be used as a "diagnostic" or for "the development of a therapeutic tool" (although the authors do not show that those cells are cytotoxic or affect PD progression).

Contribution to the field

Similar to previous studies that had suggested the involvement of T cells in the pathology of PD, Capelle et. al. identified an increase in a specific cytotoxic CD8 population in the blood of early-to-mid-stage patients compared to healthy control. These findings are of interest as little is known about the underlying mechanism of idiopathic Parkinson's disease. However, these studies are very limited in scope and utilize only FACS/CyTOF measurements. In-depth profiling of the cells (e.g., RNAseq) could shed more light on the different "programs" activated in those of cells or functional analysis (e.g. CD8-mediated cytotoxicity and exploring the role of TEMRA CD8 T-cells in PD models) were not performed. Moreover, it is not clear and wasn't discussed nor analyzed whether the changes in the immune cell populations in the blood can also be detected in the brain of idiopathic Parkinson's patients. The manuscript is quite descriptive, lack a mechanistic/functional investigation, and some of the conclusions in the discussion section are not based on experimental data. Therefore, I cannot accept the manuscript in its current form

Reply: Thanks for making these constructive comments. In order to gain additional functional and mechanistic insights, during the revision period, we performed two critical new experiments. One is the direct measurement of various cytotoxic markers (GZMA, GZMB, GZMK and Perforin) with available antibodies within different CD8 T subsets. Highly interestingly, we observed a clear enhancement in the co-expression of several cytotoxic molecules in total CD8 (especially, in CD8 TEMRA and CD8 TEM) (new **Figure 5I**). As the

synergistic effects between relevant cytotoxic molecules might be required to perform more effective cytotoxic functions, our observations further indicate that the enhanced cytotoxic capacity of CD8 T cells, especially of CD8 TEMRA and CD8 TEM. Our major observation being related to CD8 TEMRA, a population which only exists in humans, but unfortunately not mirrored in rodent models, we cannot easily perform any *in-vivo* interventional experiments to demonstrate the contribution of CD8 TEMRA to the pathogenesis of PD. In theory, one could co-culture the human iPSC-differentiated dopaminergic neurons and **autologous** CD8 T cells. However, practically speaking, it would take too much time to realize this ambitious objective. It would require first ethical approval, then additional and new biosampling to have blood and fibroblasts from the same participants, re-consenting of participants, reprogramming and differentiation of iPSC, quality control, co-culturing, experimental analysis and so on. If everything goes well, these steps alone might take at least 15 months, which is out of the scope of this work.

In the meantime, as suggested by the reviewer, we performed state-of-the-art single-cell RNA-seq analysis (scRNA-seq) on four sorted CD8 T-cell subsets (naïve, CD8 TCM, TEM, TEMRA). The reason why we had to first sort four CD8 T subsets before performing scRNA-seq analysis is that the key distinguishing markers CD45RA or CD45RO isoforms were encoded by the same gene, which cannot be distinguished by the standard widely-used Illumina shotgun sequencing methods. To our best knowledge, we are not aware that any other published scRNA-seq analysis regarding Parkinson's disease (and possibly many other diseases) has made such an effort to persuasively distinguish TEMRA from other cell (sub)-types. Our extra efforts allow us to made more convincing conclusions. These efforts also help us to better understand the potential disturbed pathways underlying our observations within CD8 (sub)-compartments. We now added several long sections in the Results in page 12-18 and in the Discussions (line 623-642, page 18-19; line 648-661, page 19) during revision, by providing two main figures for scRNA-seq results, i.e., **Figure 6** and **Figure 7**, as well as two Supplementary Figures, **Fig S6** and **Fig S7**. Essentially, we observed various enhanced cytotoxic pathways and lymphocyte trans-endothelial migration and adhesion pathways in CD8 TEMRA (as well as CD8 TEM) of early-to-mid iPD vs HC. Moreover, we also already observed more-active transcriptionally-reprogrammed CD8 TCM and naïve T cells in iPD relative to HC. The over-active status in early-differentiated CD8 T subsets might favour CD8 terminal differentiation in iPD. It is also worth to highlight: not only did we observe a substantially enhanced ratio between CD8 TEMRA and TCM in iPD vs HC by cytometry analysis, but also our unbiased pseudotime differentiation trajectory analysis using scRNA-seq clearly demonstrated an accelerated differentiation process within CD8 compartments of iPD.

As for the reported reduced CD8 Treg, we checked the literature again and indeed found one relevant paper (<https://pubmed.ncbi.nlm.nih.gov/31711508/>). Interestingly, they observed a reduction in almost every known regulatory populations, including CD4 Tregs, CD45RO+ Treg, Tr1, IL-10 producing CD8 T cells and tolerogenic PD-L1+ dendritic cells. In this context, there exist several aspects to be highlighted. First, being different from what they observed, we did not observe a reduction in the frequency of total CD4 Tregs. Second, they observed reduced IL-10-producing CD8 T cells (although they have also analysed FOXP3+ CD8 regulatory T cells, they did not observe any significant difference between PD and HC in their work). More importantly, they did not control for several critical confounding factors, e.g., immunosuppressive medications, autoimmune diseases, cancer, CMV seropositivity and others. They also did not perform gender-specific analyses. However, we controlled all these various well-known confounding factors, and performed gender-specific analysis. Therefore, by no means are their observations so conclusive and comparable to what we have generated here. We have also discussed this aspect in line 691-695, page 20.

In short, in the last year we have made substantial efforts to particularly address the major concern of this reviewer.

Minor points

1. Line 270- “..the unaffected expression levels of the analyzed chemokine receptors and major brain homing factor among CD8 TEMRA..” – in supplementary fig S4C it looks like there is a decrease in all three chemokine receptors in PD when looking at CD8 TEMRA.

Reply: Thanks for the careful reading. With a zoom-in, the chemokine receptors indeed seemed to be a bit decreased in CD8 TEMRA from the heatmap. We now carefully checked our raw data again. The percentages of CCR4+ cells among CD8 TEMRA were indeed reduced from $9.52 \pm 6.69\%$ in HC to $6.14 \pm 4.65\%$ in iPD; The percentages of CXCR3+ cells among CD8 TEMRA were reduced a bit from $2.85 \pm 2.90\%$ in HC to $1.18 \pm 1.31\%$ in iPD; The frequency of CCR6+ cells among CD8 TEMRA was indeed reduced a bit from $1.65 \pm 2.36\%$ in HC to $1.26 \pm 0.84\%$ in iPD. We believe that it is not worth to further discuss such low expression levels of CXCR3 and CCR6. Even considering CCR4, the expression of which is slightly higher compared with the other receptors, the mean of the percentages of CCR4+ cells was the lowest in CD8 TEMRA compared to that in naïve CD8, CD8 TCM and CD8 TEM with the averaged percentages of 26.52%, 54.65% and 19.93% in HC, respectively. Furthermore, if one looks at the general expression levels of those chemokine receptors across different CD8 subsets as shown in the heatmap, their expression levels in CD8 TEMRA compared with those in other CD8 memory subsets were extremely low (actually almost white in the heatmap). To avoid

misleading readers, we decided to show the overall expression patterns throughout different subsets using heatmap and concluded there is no clear difference in those chemokine receptor expression among CD8 TEMRA between PD and HC. If we used scatter dot plots showing the frequency of those chemokine receptors in CD8 TEMRA, it might lead to a wrong conclusion of “decreased expression in PD”, although the decrease occurred only in the background ‘noise’ levels and might not necessarily have any biological relevance. We now also added one sentence in line 284-286, page 9 to make this point clear.

2. Figure 4I- higher granzyme A levels- is the significance attributed to the outlier patient? Are there any other effector molecules to be tested?

Reply: Thanks for pointing out this critical point. After removing this outlier patient (value of 75.24), it is still statistical significant [please compare the left panel without the outlier value, and the right plot with the outlier point in the Figure below]. In fact, with the default setting of ROUT analysis in Graphpad, this point was indeed identified as an outlier. In the validation analysis, we further analysed cytotoxic effector molecules directly within various CD8 subsets. As shown in the new **Figure 5 E-L**, Granzyme B were enhanced, especially the coexpression of GZMA, GZMB and PRF1 was even substantially enhanced in PD vs HC, further indicating the enhanced cytotoxic capacity of CD8 TEMRA. We also analysed Granzyme K (GZMK), which expresses in exhausted-like T cells and increases during the natural ageing process (<https://www.sciencedirect.com/science/article/pii/S1074761320304921?via%3DiHub>).

Interestingly, the expression of GZMK was decreased in PD, possibly further indicating the more-active functional status of CD8 T cells in PD. In line with the cytometry data, our unbiased scRNA-seq data further demonstrated enhanced cytotoxic pathways in CD8 TEMRA of iPD vs HC.

3. Figure S5k- no significant changes in cytokines levels in the blood. If these effector cells enter the CNS, what is the changes in cytokine levels in the CSF?

Reply: Thanks for raising this intriguing point. According to a recent study (<https://www.ncbi.nlm.nih.gov/pmc/articles/PMC8629119/>), the CSF and peripheral cytokine levels are not closely correlated. As CD8 TEMRA can secrete IFN- γ and TNF- α , we speculate that IFN- γ and/or TNF- α might be enhanced in the CSF. Highly encouragingly, TNF- α has been long reported increased in both the brain and CSF of PD patients (<https://www.sciencedirect.com/science/article/abs/pii/0304394094907463>). Later on, higher levels of TNF- α has been confirmed by an independent study in the CSF of PD (<https://www.ncbi.nlm.nih.gov/pmc/articles/PMC5563061/>). We have now also discussed this point in line 654-661, page 19.

4. Didn't understand how the CMV criteria was implemented in the study. Seems that maybe CMV+ patients should be excluded.

Reply: Thanks for pointing out this issue. We indeed did not make this point clear enough in the original submission. We now tried to make this aspect clearer in the *Cohort Design* in line 794-802, page 23. There are two layers of reasons for us to focus on CMV seropositive participants. First, from the pure logistic point of view, we could not exclude the chance that some CMV seronegative participants might become CMV seropositive during the period between the previous serum sampling time and the current fresh blood sampling time. Of note,

we had to first test CMV status using available frozen sera during the previous visits from more participants to allow us to select and decide who to be invited for fresh blood sampling.

Second, according to a large-scale CMV seropositive investigation in the neighbour country (i.e., Germany) (<https://pubmed.ncbi.nlm.nih.gov/30044826/>), as it is written in their paper “Seroprevalence increased with age: from 31.8% to 63.7% in men and from 44.1% to 77.6% in women when comparing the 18-29 with the 70-79 year age-group, respectively. ”. Since German and Luxembourgish population likely share certain similarity in terms of epidemiology, we want to analyse the more-representative group of the patient/control cohort, who are already in their sixties. Choosing CMV seronegative individuals at that age range would have further severely limited the availability of suitable samples.

5. None-specific definitions such as “have shown changes” (line 70) should be avoided.

Reply: Thanks for finding this vague expression. We now have changed to a more specific statement “Not only total CD4 T cells, but also specific CD4 subsets, such as CD4 regulatory T cells (Treg) and Th17 have shown a reduction (Kustrimovic et al. 2018), although Th17 have been observed enhanced in PD patients by another study (Sommer et al. 2018)” In line 79-81, page 3.

6. Matheoud et al. 2019 do not demonstrate the “cytotoxic CD8 T-cell response against mitochondrial antigens caused PD-like motor symptoms” (lines 79—80) but merely that Gram-negative bacteria in Pink1^{-/-} mice engages mitochondrial antigen presentation and the establishment of cytotoxic mitochondria-specific CD8⁺ T cells in the periphery and in the brain.

Reply: Thanks for finding this imprecise description. We now changed it to “In a genetic-PD mouse model, mitochondria-antigen-specific CD8 T-cell responses have been shown in both the periphery and the brain” in line 90-91, page 3.

7. Fig 2C – it is not clear how the cell identifies were defined.

Reply: Thanks for helping us to identify this missing information. In the revised legend of new **Figure 1C** (Previously, it was called Figure 2C; following the suggestion of other reviewers, we now removed the previous Figure 1), we now described the marker combinations for each of the highlighted subsets. Furthermore, in **Fig. S1**, we have shown all the detailed gating strategies for different subsets analysed by CyTOF.

8. Subject criteria- patients were characterized with early-to-mid-stage up to 10 years from diagnosis. 1 patient grade 3 was included and three patients diagnosed over than 10 years. Such patients do not

follow the criteria and might alter reported results.

Reply: Thanks for identifying this inconsistency. Following our internal discussion, we decided to remove the inclusion criteria “disease duration <10 years” in the Figures, tables and text. In any case, the definition of “early-to-mid” essentially refers to the H&Y staging scale. In this way, we don’t have to re-do all these complicated analyses while still being able to remove this inconsistency. In fact, one of the three genetic patients already have disease duration of 13 years, who was included on purpose just to see whether there is any obvious difference in immunological features between idiopathic and genetic PD. Furthermore, showing the potential correlation between very narrow disease duration and CD8 TEMRA frequency is also not ideal.

We did not remove five patients with H&Y value of 3, considering two factors: i) PD patients with H&Y factor of 3 are actually still considered at middle stage (<https://parkinsonsblog.stanford.edu/2021/09/the-parkinsons-journey-understanding-progression-webinar-notes/>); ii) the already-small number of patients we selected and analysed. Further reducing the cohort sample size will not help to increase our analysis power.

9. Supplementary fig S4- different abbreviations are used for chemokine receptors in the figure and text which is confusing.

Reply: Thanks for notifying this. We have now corrected them and ensured the same abbreviations used consistently through the text and Figures.

10. Analyzing only the female population increased AUC score in ROC analysis. Will separation of CD8 TEMRA and CD8 Treg or using TEMRA/Treg ratio only on the female population will produce a better diagnostic criteria?

Reply: Thanks for pointing out this critical point. To address this, we have now performed several additional analyses only selecting female participants (see new **Figure 4 H, 4J, 4M**). Indeed, those ROC analyses based on female participants alone produce even more promising diagnostic values than analyses based on both genders. We would like to highlight that during our revision, we actually found it is the ratio between CD8 TEMRA and CD8 TCM that exhibited the highest diagnostic value (the ROC value can even reach as high as 0.9470 for those patients diagnosed within 5 years). Thanks again for making this constructive suggestion.

11. Looking at granzymes levels and other effector characteristics will be significant when analyzing only females?

Reply: Thanks for pointing out this. We have now performed gender-specific analysis on different immune subsets, e.g., CD8 TEMRA, CD8 TCM (and the ratios between CD8 TEMRA and TCM), CD8 Treg (and the ratios between CD8 TEMRA and Treg), CD8 NKT, GZMA, Neutrophils, Eosinophils and ILC2. We have now shown them in **Figure 1H, Figure 2C, Figure 2J, Figure 3B, 3D, Figure 4J, 4L-N** and **Fig. S2L, S2M, S2Q, Fig. S3D**. Of note, to reduce workload, we mainly re-examined those already showing highly-interesting results in the gender-mixed analyses. Unexpectedly, we indeed found a gender-biased observation on the indicated subsets or effector molecules. As suggested by the reviewer, we also selectively checked other effector molecules, such as serological GZMB, Perforin and others, but we did not notice significant results when analysing female-only samples (with no significance, we did not show them in this already-long paper). Pay attention, to avoid confusing the readers, we also did not show the gender-separated results of CD8 TEMRA/TCM for both CyTOF analysis and flow-cytometry analysis while only showing the corresponding flow-cytometry results in **Figure 2C** and **Fig. S3D**. We amended those gender-biased results in the corresponding places of the text (marked in red). We also discussed these gender-biased results in **line 705-714, page 20-21** and even updated the title and Abstract, as also required by the editor.

12. Line 372- written TEMTA instead of TEMRA.

Reply: We have now corrected this typo.

13. Discussion- CD45RA is not expressed solely by CD8 TEMRA (on naïve CD4 for example). Is there a way to “easily target” CD8 TEMRA as stated in the discussion? Or is it needed to be further investigated, therefore we still do not know if it really is an “easy” target.

Reply: Thanks for pointing out this confusing point. We completely agree with the reviewer on the point that CD45A is also expressed in naïve T cells. In fact, we previously used the term “easily target” mainly because of the fact that CD8 TEMRA already abundantly exist in periphery, which will make them accessible and targetable in the peripheral bloodstream. This simply means that targeting CD8 TEMRA is relatively easier than targeting something else in the brain. We now changed the term to a more precise term “periphery-accessible” in the Discussion **line 662, page 19**. We also removed the term “easily-accessible” from the Abstract.

REVIEWERS' COMMENTS

Reviewer #1 (Remarks to the Author):

In the revised version of the manuscript, the authors have adequately addressed the comments made by myself and other reviewers. The revised manuscript text and figures have been substantially improved. Collectively, there are valuable implications from the results of this study.

Reviewer #2 (Remarks to the Author):

Capelle et al. detail an interesting investigation in the immune profile of individuals with PD compared to age-matched controls. They have paid close attention to their cohort design and have accounted for many potential confounding factors (immune-altering ailments, prescription medications, etc.). On top of their stringent cohort design, Capelle et al. have embraced quality methodology in sample acquisition with the use of fresh whole blood/PBMC specimens and dual CyTOF and high-parameter flow cytometry to immunoprofile PD patient leukocytes. Their finding of an enriched CD8 TEMRA population, by both mass and flow cytometry is novel and a timely discovery in the field of neuroimmunology in PD. The authors have overall provided appropriate responses to my original critiques and have now included further evidence with their single-cell RNA sequencing experiments better detailing T cell memory populations within PD

Reviewer #3 (Remarks to the Author):

I'm delighted to recommend the acceptance of Manuscript NCOMMS-22-27239A. The authors have not only satisfactorily addressed all of my concerns and suggestions but have also demonstrated a commendable level of dedication and effort in enhancing the quality of their work.

Their responsiveness to the review process is commendable and reflects their commitment to producing high-quality scientific contributions. I am thoroughly impressed with the improvements that have significantly strengthened the manuscript.

With the revisions in place, I have full confidence in the readiness of this manuscript for publication. I enthusiastically recommend its acceptance to Nature Communications.